

# Foreign influences on tropospheric ozone over East Asia through global atmospheric transport

Han Han[1], Jane Liu[1,2,*], Huiling Yuan[1], Tijian Wang[1], Bingliang Zhuang[1], Xun Zhang[1,3]

[1]School of Atmospheric Sciences, Nanjing University, Nanjing, China

[2]Department of Geography and Planning, University of Toronto, Toronto, Canada

[3]International Institute for Earth System Science, Nanjing University, Nanjing, China

*Correspondence to: Jane Liu (janejj.liu@utoronto.ca)

**Abstract**

Tropospheric ozone in East Asia is influenced by the transport of ozone from foreign regions around the world. However, the magnitudes and variations of such influences remain unclear. This study was performed to investigate this influence and its variations with space and time using a global chemical transport model, GEOS-Chem, for emission zero-out and tagged ozone simulations. The results show that foreign ozone varies significantly with latitude, altitude, and season in the East Asian troposphere. The transport of foreign ozone to East Asia occurs primarily through the middle and upper troposphere, where the concentration of foreign ozone (32-65 ppbv) in East Asia is 0.5-6 times higher than that of native ozone (11-18 ppbv) and has strong seasonality, being largest in spring and lowest in winter. Foreign ozone in East Asia increases rapidly with altitude. At the surface, the annual average foreign ozone concentration is ~22.2 ppbv, which is comparable to its native counterpart of ~20.4 ppbv. The annual mean concentration of anthropogenic ozone from foreign regions is ~4.7 ppbv at the East Asian surface, and half of it comes from North America (1.3 ppbv) and Europe (1.0 ppbv). The presence of



foreign ozone at the East Asian surface is highest in winter (27.1 ppbv) and lowest in summer (16.5

ppbv). This strong seasonality is largely modulated by the East Asian monsoon (EAM) via its influence

on vertical motion. The large-scale subsidence prevailing during the East Asian winter monsoon

(EAWM) favours the downdraft of foreign ozone to the surface, while widespread convection in the

East Asian summer monsoon (EASM) blocks such transport. In summer, the South Asian High

facilitates the build-up of South Asian ozone in the East Asian upper troposphere and constrains North

American, European, and African ozone to the regions north of 35°N. The interannual variations of

foreign ozone at the East Asian surface have been found to be closely related to the EAM. When the

EAWM is strong, North American and European ozone are enhanced at the East Asian surface, as the

subsidence behind the East Asian trough becomes stronger. In strong EASM years, South and Southeast

Asian ozone is reduced at the East Asian surface due to weakened south-westerly monsoon wind. This

study suggests substantial foreign influences on tropospheric ozone in East Asia and underscores the

importance of the EAM in the seasonal and interannual variations of foreign influences on surface

ozone in East Asia.

## 1 Introduction

Tropospheric ozone is a major pollutant, atmospheric oxidant, and greenhouse gas (Monks et al., 2015).

Its sources include photochemical production in the troposphere and downward transport of ozone from

the stratosphere (Lelieveld and Dentener, 2000; Gettelman et al., 2011). Having a lifetime of weeks to

months in the free troposphere, ozone can be transported across regions and continents, driven by

atmospheric circulation (HTAP, 2010). Therefore, tropospheric ozone in a region is affected by both

native and foreign emissions and various physical and chemical processes at different temporal and

spatial scales (Doherty et al., 2017; Huang et al., 2017; Han et al., 2018). Atmospheric transport makes

ozone pollution a globalized issue related to health (Liang et al., 2018), climate (B. Li et al., 2016),



economics (Lin et al., 2014), and the ecosystem (Zhang et al., 2017). In recent years, East Asia has

experienced severe ozone pollution, and surface ozone concentrations are increasing (Gaudel et al.,

2018; Lu et al., 2018). Through trans-Pacific and trans-Atlantic transport, ozone precursors emitted or

ozone produced in East Asia can affect the ozone levels in North America (Verstraeten et al., 2015;

Dunker et al., 2017; Nopmongcol et al., 2017) and Europe (Karamchandani et al., 2017; Knowland et

al., 2017; Jonson et al., 2018). Nevertheless, ozone transport from foreign regions into the East Asian

troposphere has received relatively less attention.


Previous studies have assessed foreign contributions to surface ozone in East Asia by two types of

numerical simulations: emission zero-out (also known as brute-force or sensitivity, Fiore et al., 2009)

and tagged ozone simulations (Liu et al., 2011). The emission zero-out simulation examines how ozone

within a receptor region responds to perturbation of the ozone precursor emissions in different foreign

regions, while the tagged ozone simulation tracks ozone produced in separate source regions along its

transport into a receptor region. Based on the simulations, some studies have revealed that the

concentration of foreign ozone at the East Asian surface is larger in colder seasons (November-April)

than in warmer seasons (May-October) (Fiore et al., 2009; Nagashima et al., 2010; Wang et al., 2011;

Yoshitomi et al., 2011). They also found an uneven distribution of foreign ozone at the East Asian

surface (Hou et al., 2014; Y. Zhu et al., 2017; Han et al., 2018) and assessed the anthropogenic impacts

from individual source regions (Ni et al., 2018). For example, several studies suggested that 1-3 ppbv

and ~1 ppbv of surface ozone in East Asia in spring can be attributed to European (Holloway et al.,

2008; X. Li et al., 2014) and South Asian (Chakraborty et al., 2015) anthropogenic emissions,

respectively.


Concerning air quality, previous studies were mostly focused on foreign influences on the surface-





layer in East Asia. How foreign ozone is distributed in the East Asian middle and upper troposphere, where ozone has larger radiative forcing than at the lower layers (Worden et al., 2008), remains unclear (Liu et al., 2002; Sudo and Akimoto, 2007). However, limited studies have suggested that foreign

influences are much larger in the higher altitudes than at the surface in East Asia, as shown in Y. Zhu et al. (2017) and Han et al. (2018) for North American and African ozone, respectively. The contribution of anthropogenic emissions from foreign regions to ozone over China is also larger at high altitudes than at the surface in spring (Ni et al., 2018). The strongly latitude- and altitude-dependent radiative forcing of tropospheric ozone requires further examination of the vertical variation of foreign ozone in East

Asia. Quantifying foreign ozone sources at different altitudes also helps in understanding the transport mechanisms.

The transport of foreign ozone to East Asia is associated with various factors, including emissions, meteorology, and chemistry in the source regions, along the transport pathways, and in East Asia (Han

et al., 2018). Although previous studies have not fully clarified the drivers of the seasonal variation of foreign influences, they have specifically suggested that the East Asian monsoon (EAM), a predominant climate feature in East Asia, could be a key player (Wang et al., 2011; Chakraborty et al., 2015; B. Zhu et al., 2016; Y. Zhu et al., 2017; Han et al., 2018). Wang et al. (2011) suggested that the seasonal switch in wind patterns of the EAM can bring foreign ozone from different regions. Y. Zhu et al. (2017) and

Han et al. (2018) demonstrated the importance of vertical transport in the EAM. They found that the East Asian winter monsoon (EAWM) can boost downdrafts of North American ozone to the East Asian surface (Y. Zhu et al., 2017), while the prevailing convections during the East Asian summer monsoon (EASM) block such transport of African ozone (Han et al., 2018). Therefore, for better understanding of the transport of foreign ozone to East Asia, the role of EAM needs to be further investigated.




On the decadal scale, foreign ozone has been found to significantly drive the interannual variations in tropospheric ozone over East Asia (Chatani and Sudo, 2011; Sekiya and Sudo, 2012) and even lead to increasing or decreasing trends over some regions of East Asia (Nagashima et al., 2017). Y. Zhu et al. (2017) and Han et al. (2018) found that interannual variations in the transport of North American and African ozone to East Asia are closely related to the variation of the EAM intensity. However, factors controlling the interannual variations of foreign ozone over East Asia have not been adequately studied.

Since the 2000s, our understanding of the foreign influence on tropospheric ozone in East Asia has been advanced. However, previous studies individually focused on some specific aspects of this influence, such as those from one or a few source regions (X. Li et al., 2014; Y. Zhu et al., 2017; Han et al., 2018), those occurring during one or a few seasons (Ni et al., 2018), or those affecting surface ozone only in East Asia (Wang et al., 2011). The sources of ozone over the entire East Asian troposphere and the underlying transport mechanisms are not well documented. The anthropogenic and natural influences have not been separately assessed. The interannual variations of foreign influences, their sensitivity and the associated meteorology have been inadequately studied. It is desirable to systematically examine the foreign influences on tropospheric ozone in East Asia according to space and time, as well as by source region. In this study, we use a global chemical transport model, GEOS-Chem, to quantify foreign influences on tropospheric ozone in East Asia from the perspectives of anthropogenic emissions and all emissions of ozone precursors. We characterize the seasonal, latitudinal, and vertical variations of these influences and explore the potential mechanisms. We also search for a link between the interannual variations of the foreign ozone influences and the EAM. In the following portions of this paper, section 2 describes the GEOS-Chem model and the simulation experiments. The seasonal and interannual variations of the foreign influences are presented in sections 3 and 4, respectively. A summary is provided in section 5.




## 2 Model description and simulation experiments

A global three-dimensional chemical transport model, GEOS-Chem (version v9-02) (Bey et al., 2001,

http://geos-chem.org), was used to simulate the global tropospheric ozone and the transport of foreign

ozone to East Asia from different source regions. GEOS-Chem includes detailed tropospheric $O_3$-$NO_x$-

hydrocarbon and aerosol chemistry. The simulations were driven by the GEOS-4 meteorology from the

Goddard Earth Observing System (GEOS) at the NASA Global Modeling and Assimilation Office

(GMAO), with 30 reduced vertical layers at the horizontal resolution of 4° latitude by 5° longitude.

GEOS-Chem was run in two modes: the full chemistry and tagged ozone modes, corresponding to the

emission zero-out and tagged tracer simulations. The former and the latter simulate anthropogenic

ozone and overall ozone from foreign regions, respectively. Table 1 describes the experiments

conducted in this study. We divided the world into eight regions (Figure 1), including East Asia (EAS,

95°E-150°E, 20°N-60°N), North America (NAM, 170°W-65°W, 15°N-70°N), Europe (EUR, 15°E-

50°E, 35°N-70°N), Africa (AFR, 20°W-30°E, 0-35°N and 20°W-55°E, 35°S-0), central Asia (CAS,

50°E-95°E, 35°N-60°N), South Asia (SAS, 60°E-95°E, 5°N-35°N), Southeast Asia (SEAS, 95°E-

140°E, 10°S-20°N), and the rest of the world (ROW). Ten simulations in full chemistry mode were

conducted from January 2004 to February 2006 (2004 for spin-up), including one control experiment

(*CTRL*) and nine sensitivity experiments. In the *CTRL* experiment, all anthropogenic and natural

emissions were turned on, while in the sensitivity experiments, the anthropogenic emissions including

nitrogen oxides ($NO_x$), carbon monoxide (CO), and non-methane volatile organic compounds

(NMVOC) were turned off individually in each of the defined source regions and the rest of the world.

As ozone does not linearly respond to the reduction of its precursors (Fiore et al., 2009), to isolate the

relative contributions of anthropogenic emissions from different source regions to the total



anthropogenic ozone, a 'normalized marginal' linearization method (B. Li et al., 2016; Ni et al., 2018)

was used to adjust the simulations:

$$CON\text{-}A = \frac{CTRL \text{ - } EAnth\text{-}A}{\sum_{i=1}^{8} (CTRL \text{ - } EAnth\text{-}X_i)} \times (CTRL \text{ - } EAnth\text{-}GLO) \tag{1}$$

where *CON-A* is the ozone contributed by the anthropogenic emissions from a specific source region *A*.

*EAnth-Xi* indicates one of the sensitivity experiments. The difference between *CTRL* and *EAnth-GLO*

represents the total anthropogenic ozone. The calculations were conducted at every model grid.

Meteorology can modulate foreign ozone over East Asia interannually through its influences on both

transport and chemical processes (Liu et al., 2011; Sekiya and Sudo, 2012, 2014). In this study, we

focus on its impact on interannual transport. Therefore, we conducted a tagged ozone simulation from

December 1985 to November 2006 (the first year was for spin-up). In the simulation, the meteorology

was allowed to vary interannually, while there was no year-to-year variation in chemistry. We label this

simulation as Fix-Chem in Table 1. Note that seasonal variation in chemistry was allowed in the

simulation. Daily ozone production and loss data in 2005 were extracted from the full chemistry

simulation (CTRL). Then, Fix-Chem was conducted using the achieved ozone production and loss data.

Fix-Chem included 10 tracers, i.e., ozone, ozone from the stratosphere (STR), and ozone produced in

the troposphere over East Asia and over the seven foreign regions (Figure 1). A linearized ozone

parameterization scheme (Linoz, McLinden et al., 2000) was used in the calculation of stratospheric

ozone, yielding a global cross-tropopause ozone of 484 Tg in 2005. The natural and anthropogenic

emissions of ozone precursors and the model configuration were the same as those in Han et al. (2018)

and are described in detail there.

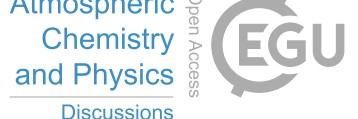

To assess the sensitivity of the GEOS-Chem simulation to different meteorological data, spatial

resolutions, and simulation modes, we ran GEOS-Chem driven by both GEOS-4 and GEOS-5 at two

spatial resolutions (4° by 5° and 2° by 2.5° in latitude and longitude) in full chemistry mode and using

GEOS-4 data at 4° by 5° resolution in tagged ozone modes. Figure 2 compares the ozone vertical

profiles from these simulations, averaged over East Asia by season. Among the different meteorological

data and resolutions, the simulated ozone profiles over East Asia were similar in shape and magnitude in

every season, except near the surface, where the simulated ozone concentrations obtained with GEOS-4

data were larger than those obtained with GEOS-5. The differences between these simulations were

smaller in summer and autumn than in winter and spring. A difference within ±5% existed between the

tagged ozone and full chemistry simulations, likely owing to the nonlinearity in chemistry. We further

compared the GEOS-Chem simulations with the ozone retrievals from to the Tropospheric Emission

Spectrometer (TES) using the monthly product TL2O3LN achieved from NASA Langley Atmospheric

Science Data Center (https://eosweb.larc.nasa.gov/project/tes/tes_table). The GEOS-Chem simulations

smoothed with TES *a priori* and the averaging kernels appeared lower than the TES measurements in

the middle troposphere by approximately 10 ppbv in spring and 5 ppbv in the other seasons (Figure 2).

Note that TES tropospheric ozone retrievals generally have a positive bias compared with ozonesonde

measurements (Nassar et al., 2008; Verstraeten et al., 2013). Verstraeten et al. (2013) identified that the

bias is approximately 2-7 ppbv and is different for the tropics (3 ppbv), sub-tropics (5 ppbv), and mid-

latitudes (7 ppbv). Our confidence in the GEOS-Chem performance is also based on extensive

validation of GEOS-Chem simulations of tropospheric ozone in East Asia (Wang et al., 2011; Jiang et

al., 2015; J. Zhu et al., 2017; Y. Zhu et al., 2017), North America (Zhang et al., 2008; Y. Zhu et al.,

2017), Europe (Liu et al., 2005; Kim et al., 2015), Africa (Han et al., 2018), and other regions (Liu et

al., 2009; Jiang et al., 2016).



To clearly identify different sources of tropospheric ozone in East Asia, we define some terms used in this paper (Table 2). East Asia is the receptor region (Figure 1), and the regions outside East Asia are the source or foreign regions. Ozone in this paper refers to ozone in the troposphere, unless stated

otherwise. Tropospheric ozone in East Asia consists of ozone produced in the troposphere from East Asia and foreign regions and ozone transported to the troposphere from the stratosphere. In this paper, we discuss the foreign influence on tropospheric ozone in East Asia from multiple perspectives (Table 2). (1)The first is by regions inside and outside East Asia, in which the terms "native ozone" and "foreign ozone" refer to ozone produced in the troposphere inside and outside East Asia, respectively.

"Foreign ozone", in this context, generally refers to ozone originally produced in foreign regions and distributed in the domain of East Asia. It also can refer to ozone originally produced in foreign regions and distributed outside East Asia, depending on the context. (2) The second is by foreign region, in which ozone produced in the troposphere over a foreign region is named after that region, such as "North American ozone". (3) The third is by the sources of ozone precursors, in which the term

"anthropogenic ozone" refers to ozone produced from precursors with anthropogenic sources. "Natural ozone" is the sum of ozone produced in the troposphere from precursors with natural sources and ozone from the stratosphere. (4) In the fourth discussion perspective, the components of ozone are termed in more detail by further divisions, such as by both sources of ozone precursors and foreign regions, as defined in Table 2. For example, North American anthropogenic ozone refers to ozone produced from

anthropogenic precursors emitted from North America. The foreign influence is assessed in terms of the absolute contribution with a unit of ppbv and the fractional contribution with a unit of percentage (%), which is the ratio of foreign ozone to ozone in the same domain of interest, unless stated otherwise.



## 3 Seasonality of foreign ozone over East Asia

### 3.1 Native and foreign ozone over East Asia

Based on GEOS-Chem simulations in 2005, we show the seasonal variations of native and foreign ozone over East Asia at the surface (Tables 3 and 4) and in the East Asian troposphere (Figures 3 and 4). The tagged ozone simulation evaluates the amount of foreign ozone generated from both anthropogenic and natural precursors, as well as the amount of ozone from the stratosphere (Figure 3, Table 3), while the emission zero-out simulation further specifies the amounts of anthropogenic ozone from foreign regions (Figure 4, Table 4).

At the East Asian surface, the annual mean foreign ozone concentration is ~22.2 ppbv, which is comparable to that of its native counterpart (~20.4 ppbv) (Table 3). Seasonally, foreign ozone is the largest in winter (27.1 ppbv), the second largest in spring (25.4 ppbv) and the smallest in summer (16.6 ppbv) (Table 3, Figure 3). Foreign ozone accounts for over 50% of ozone at the East Asian surface throughout the year, except in summer. This is similar to the estimate of 50-80% foreign contributions in spring made by Nagashima et al. (2010) and J. Li et al. (2016). Differently, foreign anthropogenic ozone at the East Asian surface peaks in spring, at 6.4 ppbv, and accounts for 14.1% of ozone (Table 4, Figure 4). This level is comparable to that of native anthropogenic ozone in spring, i.e., a 6.9 ppbv concentrations and a 15.1% fractional contribution. These results are in agreement with Ni et al. (2018), who estimated that foreign and native anthropogenic ozone were both approximately 6 ppbv at the surface in China in 2008, as determined through a GEOS-Chem simulation. However, Wang et al. (2011) suggested a higher estimate of 14.8±2.2 ppbv foreign anthropogenic ozone in China. In summer, foreign ozone (16.5 ppbv, 35.2% of ozone) and foreign anthropogenic ozone (3.7 ppbv, 8.2%) at the East Asian surface are both at seasonal minimums, whereas native ozone (30.1 ppbv, 64.1%) and native anthropogenic ozone (9.8 ppbv, 21.8%) both reach the seasonal maximum (Tables 3 and 4, Figures 3



and 4).

240    In the middle troposphere at 500 hPa, foreign ozone (47.3 ppbv, 72% of ozone) and foreign
anthropogenic ozone (9 ppbv, 15%) both peak in spring (Figures 3 and 4). In the upper troposphere at
300 hPa, foreign ozone is highest in spring (63.4 ppbv, 63% of ozone) and lowest in winter (51.1 ppbv,
57%), while foreign anthropogenic ozone is highest in summer (9.5 ppbv, 11%) and lowest in winter
(6.3 ppbv, 7%).


The seasonal variation of foreign influences on tropospheric ozone over East Asia is modulated by
multiple factors. The chemical lifetime of ozone is one of the important factors, as a longer lifetime can
lengthen the transport distance. As shown in Figure 5, the chemical lifetime of ozone at the East Asian
surface is longest in winter (32 days), shortest in summer (3.5 days) and intermediate in spring (11
days) and autumn (10 days). The chemical lifetime of ozone is approximately 10 days longer in the
middle troposphere (500 hPa) than in the boundary layer (Figure 5). This is expected, considering dry
deposition of ozone near the surface (Fiore et al., 2002; Wang et al., 2011). From the meteorological
perspective, subtropical westerlies are the major transport pathway for atmospheric pollutants moving
from the west to East Asia (Wild et al., 2004; Y. Zhu et al., 2017; Han et al., 2018). The strength and
location of the westerlies vary with season. The East Asian subtropical westerly jet is strongest in winter
and weakest in summer (Figure 6, also see Zhang et al., 2006). Therefore, the combined effect of the
ozone lifetime and the westerlies would be more favourable to the transport of foreign ozone to East
Asia in winter and spring than in summer. However, because the inflows to East Asia in winter have low
ozone concentrations (Figures 6d and 6h), both foreign ozone and foreign anthropogenic ozone in the
East Asian upper troposphere are at minimum values in winter (Figures 3 and 4).

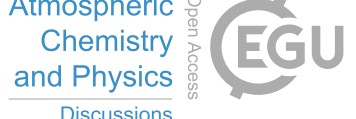



Because of the longer lifetime of ozone and the stronger westerly wind in the middle and upper troposphere (Figures 5 and 6), foreign ozone is 1-2 times higher there than at the surface (Figure 3). Figure 6 clearly shows that the transport of foreign ozone to East Asia occurs mainly through the middle

and upper troposphere. It also demonstrates that the seasonality of foreign ozone in different tropospheric layers, particularly in layers near the surface, is greatly impacted by the vertical transport. In East Asia, the downdrafts behind the East Asian trough in the EAWM favour the descent of foreign ozone from upper levels to the surface. Winter has the strongest downdrafts, followed by spring (Figure 6). Oppositely, in summer, the prevailing ascents in the EASM block foreign ozone from reaching the

lower troposphere (Figure 6b, see also Y. Zhu et al., 2017; Han et al., 2018). Combined with the obstruction of the Tibetan Plateau, the blocking effect is obvious in summer (Figures 6b and 6f, see also Han et al., 2018). In addition, the downdrafts behind the European trough (~0-40 °E in Figures 6b and 6f) divert foreign ozone from reaching East Asia in summer (Y. Zhu et al., 2017; Han et al., 2018). Because of all of the above reasons, both foreign anthropogenic ozone and foreign ozone at the surface-

layer in East Asia are higher in winter than in summer (Figures 3 and 4). Overall, the EAM is a dominant meteorological system influencing the seasonal variations of foreign ozone at the East Asian surface, primarily through the vertical motion of air masses.

Vertically, foreign ozone constantly increases with altitude during all seasons (Figures 3a-3d). The

concentration of foreign ozone is larger than that of native ozone throughout the troposphere in spring, autumn, and winter and is above ~650 hPa in summer (Figures 3a-3d). In terms of the fractional contributions, foreign ozone (Figures 3e-3h) and foreign anthropogenic ozone (Figures 4e-4h) both peak in the middle or upper troposphere, depending on the season. In contrast, native ozone, either from all sources or from anthropogenic sources, is the largest near the surface and decreases with altitude. These

large differences between foreign ozone and foreign anthropogenic ozone in the upper troposphere



imply the importance of non-anthropogenic sources at these altitudes (Figure 3 vs. Figure 4, also see Liu et al., 2002; Aghedo et al., 2007).

## 3.2 Foreign ozone over East Asia by source region

**3.2.1 Foreign ozone at the East Asian surface by source region**

In this section, we further examine the seasonality of foreign ozone over East Asia by source region. Tables 3 and 4, respectively, show foreign ozone and foreign anthropogenic ozone (in ppbv and percentage) at the East Asian surface by region. Annually, on average (Table 3), foreign ozone plus stratospheric ozone at the East Asian surface is ~23.5 ppbv (54% of ozone). The annual mean ozone

concentrations from each of the foreign regions range between 0.9-6.2 ppbv, which account for 2.0-14.2% of surface ozone. The largest contributing region is the ROW, followed by Europe, central Asia, North America, South Asia, Africa, and Southeast Asia. Seasonally, ozone from North America (5.7 ppbv, 14% of surface ozone), Europe (5.3 ppbv, 13%), Africa (2.4 ppbv, 6%), and the ROW (7.5 ppbv, 18%) peaks in winter, whereas ozone from South Asia (2.9 ppbv, 6%) peaks in spring and ozone from

Central Asia (4.7 ppbv, 10%) and Southeast Asia (1.3 ppbv, 3%) peak in summer. The seasonalities of North American, European, and African ozone are similar, with all decreasing from winter to summer and then increasing in autumn (Table 3), forming a unimodal distribution. South Asian ozone exhibits a unimodal seasonality, as well (Chakraborty et al., 2015).

By ozone precursor, each of the foreign regions contributes less than 3 ppbv of anthropogenic ozone to the East Asian surface ozone during all seasons (Table 4). On average, annually, the largest contributing regions are North America (27% of foreign anthropogenic ozone) and Europe (21%), followed by South Asia (16%), central Asia (14%), and Southeast Asia (12%). Seasonally, a springtime maximum appears for anthropogenic ozone from Europe (1.8 ppbv, 4% of surface ozone), central Asia


(1.0 ppbv, 2%), and South Asia (0.9 ppbv, 2%), while anthropogenic ozone from North America is high

in both winter (1.8 ppbv, 5%) and spring (1.7 ppbv, 4%). Anthropogenic ozone from Africa and the

ROW is the smallest at, in total, less than 1 ppbv throughout the year. It is noteworthy that ROW ozone

from anthropogenic sources contributes little to the East Asian surface ozone (0.8% of surface ozone,

Table 4), but ROW ozone from all sources contributes much more greatly (14.2%, Table 3), implying

the importance of natural emissions in the ROW to the East Asian surface ozone.

The foreign anthropogenic influence on the East Asian surface ozone in springtime was previously

studied. Fiore et al. (2009) found that if anthropogenic emissions of ozone precursors were reduced by

20% in spring over North America, Europe, and South Asia, surface ozone in East Asia would decrease

by 0.3-0.4, 0.2-0.3, and 0.1-0.2 ppbv, respectively. Chakraborty et al. (2015) suggested a decrease of 0.2

ppbv ozone at the East Asian surface in response to a 20% reduction of anthropogenic emissions in

South Asia. The results of this study appear comparable with the simulations in Fiore et al. (2009) and

Chakraborty et al. (2015), although there exist uncertainties in our results because of the chemical

nonlinearity (Huang et al., 2017). The results are also similar to those in X. Li et al. (2014), who

estimated that European and South Asian anthropogenic emissions can each contribute 2.4 and 1 ppbv

to surface ozone in western China. Holloway et al. (2008) reported that anthropogenic ozone from both

North America and Europe ranges between 1-3 ppbv in various regions in East Asia. Ni et al. (2018)

suggested that these two regions contribute 1.6 and 1.4 ppbv, respectively, to surface ozone in China in

the spring of 2008. Wild et al. (2004) demonstrated that North American and European anthropogenic

ozone in spring each range from 1.5 to 2.5 ppbv at the surface over Japan. Using the same model but

with different emission data from those in Wild et al. (2004), Yoshitomi et al. (2011) simulated that

North American and European anthropogenic ozone contribute 2.8±0.5 and 3.5±1.1 ppbv, respectively,

to surface ozone in Japan during February-April. Simulations from this study are at the same magnitude





as those in previous studies and are slightly smaller than those in Yoshitomi et al. (2011).


### 3.2.2 Foreign ozone in the East Asian middle and upper troposphere by source region

Figure 7 shows the vertical distributions of foreign ozone (in ppbv) in East Asia by region (Figures 7a-7d), along with the fractional contributions to ozone produced in the troposphere (in %) (Figures 7e-7h) and to ozone over East Asia at the corresponding altitudes (Figures 7i-7l). In the East Asian middle and

upper troposphere, foreign regions contribute to 65-85% of ozone produced in the troposphere. When foreign ozone peaks at 500 hPa in winter, the ROW is the largest contributor (13.5 ppbv, 31% of foreign ozone), followed by North America (9.1 ppbv, 21%), Africa (7.1 ppbv, 16%), Europe (4.9 ppbv, 11%), and South Asia (4.3 ppbv, 10%). At 300 hPa, the concentrations of wintertime ozone from the ROW (17.3 ppbv), North America (10.9 ppbv), and Africa (8.0 ppbv) are all higher than those of native ozone

(5.6 ppbv). The sum of the ozone from these three regions accounts for 71% of foreign ozone. Excluding the ROW, North America is the region that contributes the most ozone to the East Asian middle and upper troposphere.

Vertically, foreign ozone from five of the source regions, North America, Africa, South Asia,

Southeast Asia, and the ROW, increases obviously with altitude in the East Asian troposphere during all seasons (Figure 7). European ozone decreases with altitude in the East Asian troposphere in winter, which is opposite to the vertical variations in the other seasons. The peaking season of central Asian ozone is summer in all tropospheric layers in East Asia, and in this season, its vertical maximum occurs in the middle troposphere.


Stratospheric ozone in the East Asian troposphere increases rapidly with altitude, is largest in winter, and is the second largest in spring (Figure 7). For instance, it rises from below 3 ppbv (5% of ozone) at





the surface to 3-30 ppbv in the middle troposphere (5%-30%) and above 30 ppbv (>30%) in the upper

troposphere in spring (Figures 7a and 7i). The simulated springtime stratospheric ozone at the East

Asian surface in this study is lower than those in some previous studies that used different global

chemical transport models or different versions of GEOS-Chem, in which the stratospheric contribution

was estimated as 10%-26% by Nagashima et al. (2010), 4-10 ppbv in China by Wang et al. (2011), and

11.2±2.5 ppbv (February-April) in Japan by Yoshitomi et al. (2011). The magnitude and seasonality of

the stratospheric influence in the middle troposphere are similar to those in B. Zhu et al. (2016), who

showed that stratospheric ozone at some mountain sites (>1500 m) in China peaks in winter, at

approximately 10-15 ppbv. By comparing GEOS-Chem simulations, MLS satellite observations, and

MERRA and MERRA-2 reanalysis data, Jaeglé et al. (2017) suggested that GEOS-Chem

underestimates the ozone enhancement from stratospheric intrusions in extratropical cyclones by a

factor of 2, corresponding to a systematic underestimate of ozone in the lowermost extratropical

stratosphere.

### 3.2.3 Latitudinal variations of foreign ozone in the East Asian troposphere by source region

Figure 8 shows how the fractional contributions of foreign ozone vary with latitude at the surface and in

the middle troposphere. At the East Asian surface (Figure 8a-8d), the fractional contribution of foreign

ozone along all latitudes peaks at the northern border of East Asia (60°N) in the four seasons, ranging

from 55% in summer to 85% in winter. In spring, native ozone at the East Asian surface is largest at

approximately 30°N (36.8 ppbv, 65.8% of ozone). The latitude at which native ozone peaks shifts

northward to 35°N in summer (50 ppbv, 88% of ozone, Figure 8b) and then southward to 25°N in

autumn (31 ppbv, 74%, Figure 8c) and farther southward to 20°N in winter (28 ppbv, 61%, Figure 8d).

This seasonal migration may be partially related to the variation of ozone production influenced by the

EAM (Hou et al., 2015; S. Li et al., 2018). The concentration of stratospheric ozone at the East Asian



surface is largest between 42°N and 46°N in spring (2.4 ppbv) and winter (2.7 ppbv).

In nearly all the tropospheric layers (Figure 8e-8h, the cases in the upper troposphere are not shown), South and Southeast Asian ozone is mostly distributed in the regions south of 35°N in East Asia. In contrast, ozone from North America, Europe, and central Asia mostly appears north of 35°N in East Asia. The fractional contributions of foreign ozone from South Asia, Southeast Asia, and Africa all decrease with latitude. In contrast, the fractional contributions of ozone from North America, Europe, and central Asia all increase with latitude. The latitudinal variations of North American and European

ozone are consistent with those in Hou et al. (2014) and Y. Zhu et al. (2017).

The latitudinal variations of foreign ozone from different source regions are likely due to the proximity of these foreign regions to East Asia, the topography in East Asia, and the meteorology along the transport pathways. In particular, in summer, atmospheric circulations in the upper troposphere over

Eurasia are greatly influenced by the South Asian High (SAH, or so-called Asian summer monsoon anticyclone or Tibetan High), and its position and coverage are shown by the streamlines in Figure 9. Figure 9 illustrates that the SAH constrains foreign ozone from North America, Europe, Africa and the ROW to latitudes north of 35°N in the East Asian upper troposphere. The SAH also blocks the northward transport of Southeast Asian ozone to East Asia. Furthermore, the SAH facilitates the build-

up of South Asian ozone in the East Asian upper troposphere (Figure 9f) (Vogel et al., 2015), being a reason for the summer maximum of South Asian ozone over the region (Figure 7b).

## 4 Interannual variations of foreign ozone at the East Asian surface

The Fix-Chem simulation (Table 1) was used to search for possible connections of the interannual

variations between meteorology and foreign ozone in East Asia. The Fix-Chem simulation provided the



means and year-to-year variations of native and foreign ozone from the different regions during the 20-year period (not shown). The mean native and foreign ozone at the East Asian surface were in close agreement with those in 2005 (Table 3) in the corresponding regions and seasons.

As a typical monsoon region, the climate in East Asia is largely influenced by the EAM. The monsoon circulation can impact ozone transport and distribution in East Asia (Y. Zhu et al., 2017; S. Li et al., 2018). To search for possible linkages between the EAM and the transport of foreign ozone to East Asia, we selected three EAWM indices and three EASM indices, respectively, for winter and summer. These monsoon indices were proposed to describe the features of the EAM from different

perspectives (Q. Li et al., 2016; S. Li et al., 2018). The monsoon indices are each correlated with different types of foreign ozone at the East Asian surface. The linkages between the EAM and foreign ozone at the East Asian surface were assessed according to the mean of the correlation coefficients from the three indices in a season.

These six monsoon indices are widely used. The three EAWM indices were proposed by Sun and Li (1997), Jhun and Lee (2004), and Wang and Jiang (2004), respectively, corresponding to Equations (2)-(4). EAWMI1, defined by Sun and Li (1997), represents the EAWM strength by the averaged geopotential heights in the middle troposphere in the location of the East Asian trough, which is an important component of the EAWM. EAWMI2 (Jhun and Lee, 2004) reflects the meridional wind shear

associated with the jet stream in the upper troposphere, mainly describing the variability of the EAWM in the East Asian mid-latitudes. EAWMI3 (Wang and Jiang, 2004) uses the anomaly of the wind velocity around the coast of East Asia in the lower troposphere. The three EASM indices were proposed by Wang and Fan (1999), Li and Zeng (2002), and Zhang et al. (2003), respectively, corresponding to Equations (5)-(8). EASMI1 (Wang and Fan, 1999) is defined from the shear vorticity in the lower





troposphere that reflects variations in both the monsoon trough and the subtropical high (Wang et al.,

2008). EASMI2 (Li and Zeng, 2002) is a unified dynamical index of the monsoon which characterizes

the seasonal and interannual variability of monsoons over different areas in the world. EASMI3 (Zhang

et al., 2003) is a vorticity index similar to that in Wang and Fan (1999) but in a slightly modified

domain.


$$EAWMI1 = GPH_{500}(30\text{-}45\ ^oN,\ 125\text{-}145\ ^oE) \tag{2}$$

$$EAWMI2 = U_{300}(27.5\text{-}37.5\ ^oN,\ 110\text{-}170\ ^oE) - U_{300}(50\text{-}60\ ^oN,\ 80\text{-}140\ ^oE) \tag{3}$$

$$EAWMI3 = WS_{850}(25\text{-}50\ ^oN,\ 115\text{-}145\ ^oE) \tag{4}$$

$$EASMI1 = U_{850}(5\text{-}15\ ^oN,\ 90\text{-}130\ ^oE) - U_{850}(22.5\text{-}32.5\ ^oN,\ 110\text{-}140\ ^oE) \tag{5}$$

$$EASMI2 = \delta(10\text{-}40\ ^oN,\ 110\text{-}140\ ^oE,\ 850\ hPa) \tag{6}$$

$$\delta = \frac{\parallel \overline{V}_1 - V_i \parallel}{\parallel \overline{V} \parallel} - 2 \tag{7}$$

$$EASMI3 = U'_{850}(10\text{-}20\ ^oN,\ 100\text{-}150\ ^oE) - U'_{850}(25\text{-}35\ ^oN,\ 100\text{-}150\ ^oE) \tag{8}$$

In Equations (2)-(6), *GPH$_{500}$ is* the geopotential height at 500 hPa, $U_{300}$ is the zonal wind at 300 hPa,

*WS$_{850}$* is the wind speed at 850 hPa, $U_{850}$ is the zonal wind at 850 hPa, and $\delta$ is a dynamical normalized

seasonality obtained from Equation (7). In Equation (7), $\overline{V}_1$ and $V_i$ are the January climatological and

monthly wind vectors at a grid and $\overline{V}$ is the mean of the January and July climatological wind vectors at

the same grid. The norm $\parallel V \parallel$ is defined as $\left( \iint_s |V|^2 dS \right)^{1/2}$, where S denotes the domain of integration.

Figure 10 shows the interannual variations of foreign and native ozone at the East Asian surface driven

by meteorology and the strength of the EAM. In winter (Figure 10a), the transport of both North

American and European ozone is significantly related to the strength of the EAWM. The EAWM can



explain more than 30% of the interannual variations of ozone at the East Asian surface from these two

regions. A positive correlation between North American ozone and the EAWM was also found by Y.

Zhu et al. (2017), who suggested that the increase of ozone transport under a strong EAWM condition is

mainly caused by the enhanced downdraft from the Siberian High and the East Asian trough. This

explanation can also be applied to European ozone. In contrast, native ozone is negatively correlated

with the interannual variation of the EAWM strength. When the EAWM is strong in winter, the

enhanced Siberian High strengthens the northerly wind in the East Asian lower troposphere, and,

consequently, more native ozone is taken away (Q. Li et al., 2016).

In summer (Figure 10b), the interannual variations of the EASM strength were found to be positively

correlated with those of ozone and native ozone but negatively correlated with those of South Asian and

Southeast Asian ozone at the East Asian surface. The positive correlation between the EASM and

surface ozone in East Asia was also reported in previous studies (Zhou et al., 2013; Yang et al., 2014;

Hou et al., 2015; S. Li et al., 2018). In a strong EASM year, a clear anomalous cyclonic circulation

appears over the area southeast of China in the lower troposphere, and the south-westerly monsoon

wind is weakened, as depicted by multiple studies (for example, Figure 5 in Yang et al. (2014) and

Figure 2 in S. Li et al. (2018)). The weakened south-westerly monsoon wind during a strong EASM

year enhances ozone and native ozone at the East Asian surface by reducing ozone export (Yang et al.,

2014). Meanwhile, the south-westerly wind brings less South Asian and Southeast Asian ozone to the

East Asian surface. The variation of the EASM intensity can approximately explain 32%, 31%, and

64% of the interannual variability in native, South Asian, and Southeast Asian ozone, respectively.

**6 Discussion and conclusions**

In this numerical study using a global chemical transport model, GEOS-Chem, we investigated foreign





influences on tropospheric ozone over East Asia. We estimated these influences from the perspectives of the anthropogenic and total emissions, respectively, using the emission zero-out and tagged ozone simulations. The distributions of foreign ozone in East Asia were characterized in space (vertically and

latitudinally) and time (seasonally). Based on six EAM indices, links between the EAM and interannual variations of foreign ozone at the East Asian surface were explored. Conclusions were drawn as follows.

       The transport of foreign ozone to East Asia occurs mainly through the middle and upper troposphere because of the longer lifetime of ozone and the stronger westerlies in the northern hemisphere. In the

East Asian upper troposphere (700-200 hPa), the concentration of foreign ozone is 0.5-6 times higher than that of native ozone, with the former ranging between 32-65 ppbv (65-85% of ozone produced in the troposphere) and the latter ranging from 11-18 ppbv (Figures 3 and 7). At the surface, the annual mean concentrations of foreign ozone (22.2 ppbv) and native ozone (20.4 ppbv) are comparable (Table 3). North America and Europe are the two major contributing regions of surface ozone in East Asia.

Considering only anthropogenic ozone, the total foreign influence (4.7 ppbv, 43%) is slightly lower than its native counterpart (6.3 ppbv, 57%) in terms of the annual mean (Table 4). Three-quarters of the foreign anthropogenic ozone at the surface is from North America (27%), Europe (22%), South Asia (16%), and Southeast Asia (12%).

Foreign ozone at the East Asian surface is greatly modulated by the downward transport foreign ozone in East Asia, and thus, its seasonality is dominated by the EAM system (Figure 6). The subsidence prevailing in the EAWM favours the downward transport of foreign ozone to the surface, while ascending flows in the EASM block such transport. Therefore, foreign ozone at the East Asian surface is highest in winter (27.1 ppbv, 66% of surface ozone) and lowest in summer (16.5 ppbv, 35%)

(Table 3).



In nearly all tropospheric layers, foreign ozone from North America, Europe, and central Asia generally increases with latitude from 20°N to 60°N in East Asia, while a decrease of foreign ozone with latitude is observed for the regions of South Asia and Southeast Asia (Figure 8). In the upper troposphere, the SAH in summer blocks North America, European, and African ozone from transport to latitudes south of 35°N in East Asia (Figure 9).

The interannual variations of foreign ozone at the East Asian surface were found to be closely related to the EAM strength (Figure 10). When the EAWM is strong in winter, more North American and European ozone tends to be transported to the East Asian surface because of the heavier downdrafts behind the East Asian trough. Meanwhile, the strengthened north-westerly and north-easterly monsoon winds can reduce the native ozone by enhancing its export. When the EASM is strong in summer, the weakened south-westerly monsoon wind enhances the native contribution by decreasing ozone export from China. In the meantime, the weakened south-westerlies reduce the transport of South Asian and Southeast Asian ozone to the East Asian surface.

This study reveals the significant foreign influences on tropospheric ozone over East Asia through global atmospheric transport. While previous studies investigated ozone transport from one or a few foreign regions (X. Li et al., 2014; Chakraborty et al., 2015), that during one or a few seasons (Ni et al., 2018) or only surface ozone in East Asia (Wang et al., 2011), a comprehensive assessment is provided in this study. Our results are generally consistent with those of previous studies and appear to be reasonable. For example, we concluded that foreign ozone contributes ~50% of springtime surface ozone in East Asia, and this is in agreement with Nagashima et al. (2010) and J. Li et al. (2016). However, there are some disagreements with previous studies concerning various details. Upon



considering the foreign anthropogenic ozone influence, our estimates for North America and Europe are slightly lower than those of Yoshitomi et al. (2011), probably resulting from differences in the emission inventories and numerical models. The simulated stratospheric influence appears weaker than in some previous studies (Nagashima et al., 2010; Wang et al., 2011), possibly due to the underestimation of ozone in the lowermost stratosphere by GEOS-Chem (Jaeglé et al., 2017).


    We examined the foreign influence on ozone in East Asia throughout all tropospheric columns. Such influence in the East Asian middle and upper troposphere is important to climate change because of the considerable ozone radiative forcing over the area (Myhre et al., 2017). The simulations show that the concentration of foreign ozone increases remarkably with altitude and is much higher than its native

counterpart in the middle and upper troposphere (Figure 7), implying the significant role that foreign ozone may play in climate change in East Asia. Such an impact has been rarely documented (Sudo and Akimoto, 2007).

    In this study, anthropogenic emissions in 2005 were used, which were scaled from the most recent

inventories, such as the global inventory EDGAR v3.2 in 2000 (Olivier and Berdowski, 2001; Pulles et al., 2007), the INTEX-B Asia emissions in 2006 (Zhang et al., 2009), and the NEI05 for North America in 2005. Our simulations in spring agree with those in Ni et al. (2018), who used the global inventory EDGAR v4.2 in 2008. However, the anthropogenic emissions have significantly changed globally in the last decade, especially in East Asia. Zheng et al. (2018) suggested that because of clear air actions, the

anthropogenic emissions in China during 2010-2017 decreased by 17% for $NO_x$ and 27% for CO and increased by 11% for NMVOC. Moreover, the anthropogenic $NO_x$ and CO emissions in North America are decreasing (Jiang et al., 2018). In addition, there have been some changes in the global natural emissions of ozone precursors, such as in biogenic NMVOC emissions (Chen et al., 2018). Therefore,



updated emissions inventories can improve future estimates of foreign influences on ozone over East Asia.

Regarding the meteorological impact on the interannual variation of foreign ozone in East Asia, we specifically underscored the importance of EAM, representing an advancement from Y. Zhu et al. (2017) and Han et al. (2018). Future studies can further examine the influences of other prominent
climate systems, for instance, the North Atlantic Oscillation (NAO) (Bacer et al., 2016) and the El Niño Southern Oscillation (ENSO) (Sekiya and Sudo, 2012, 2014; Hou et al., 2016).

**Data availability**

The GEOS-Chem model is available at http://acmg.seas.harvard.edu/geos/. The TES ozone data can be
downloaded from https://eosweb.larc.nasa.gov/project/tes/tes_table.

**Author contributions**

H. Han and J. Liu designed and conducted the simulations, analysed the results, and wrote the manuscript. J. Liu conceived the research problems and supervised the study. H. Yuan, T. Wang, B.
Zhuang, and X. Zhang contributed to the data analysis and result interpretation.

**Competing interests**

The authors declare that they have no conflict of interest.

**Acknowledgements**

We gratefully acknowledge that the GEOE-Chem model has been developed and managed by the Atmospheric Chemistry Modeling Group at Harvard University. The TES ozone data were acquired



from the NASA Langley Atmospheric Science Data Center. The NCEP/NCAR reanalysis data were
from the NOAA Earth System Research Laboratory. This research is supported by the Chinese Ministry
of Science and Technology under the National Key Basic Research Development Program
(2016YFA0600204, 2014CB441203) and by the Natural Science Foundation of China (41375140,
91544230).

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

Background ozone over the United States in summer: Origin, trend, and contribution to pollution
episodes, J. Geophys. Res., 107, D15, https://doi.org/10.1029/2001JD000982, 2002.

Gaudel, A., Cooper, O., Ancellet, G., Barret, B., Boynard, A., Burrows, J., Clerbaux, C., Coheur, P.-F.,
Cuesta, J., and Cuevas Agulló, E.: Tropospheric Ozone Assessment Report: Present-day distribution





and trends of tropospheric ozone relevant to climate and global atmospheric chemistry model

evaluation, Elem. Sci. Anth., 6, 39, https://doi.org/10.1525/elementa.291, 2018.

Gettelman, A., Hoor, P., Pan, L. L., Randel, W. J., Hegglin, M. I., and Birner, T.: The extratropical upper

troposphere and lower stratosphere, Rev. Geophys., 49, RG3003,

https://doi.org/10.1029/2011RG000355, 2011.

Han, H., Liu, J., Yuan, H., Zhuang, B., Zhu, Y., Wu, Y., Yan, Y., and Ding, A.: Characteristics of

intercontinental transport of tropospheric ozone from Africa to Asia, Atmos. Chem. Phys., 18, 4251-

4276, https://doi.org/10.5194/acp-18-4251-2018, 2018.

Holloway, T., Sakurai, T., Han, Z., Ehlers, S., Spak, S. N., Horowitz, L. W., Carmichael, G. R., Streets,

D. G., Hozumi, Y., Ueda, H., Park, S. U., Fung, C., Kajino, M., Thongboonchoo, N., Engardt, M.,

Bennet, C., Hayami, H., Sartelet, K., Wang, Z., Matsuda, K., and Amann, M.: MICS-Asia II: Impact

of global emissions on regional air quality in Asia, Atmos. Environ., 42, 3543-3561,

https://doi.org/10.1016/j.atmosenv.2007.10.022, 2008.

Hou, X., Zhu, B., Fei, D., and Wang, D.: The impacts of summer monsoons on the ozone budget of the

atmospheric boundary layer of the Asia-Pacific region, Sci. Total Environ., 502, 641-649,

https://doi.org/10.1016/j.scitotenv.2014.09.075, 2015.

Hou, X., Zhu, B., Fei, D., Zhu, X., Kang, H., and Wang, D.: Simulation of tropical tropospheric ozone

variation from 1982 to 2010: The meteorological impact of two types of ENSO event, J. Geophys.

Res., 121, 9220-9236, https://doi.org/10.1002/2016JD024945, 2016.

Hou, X., Zhu, B., Kang, H., and Gao, J.: Analysis of seasonal ozone budget and spring ozone latitudinal

gradient variation in the boundary layer of the Asia-Pacific region, Atmos. Environ., 94, 734-741,

https://doi.org/10.1016/j.atmosenv.2014.06.006, 2014.

HTAP: Hemispheric transport of air pollution 2010, United Nations, edited by: Dentener, F., Keating, T.

and Akimoto, H., New York and Geneva, 2010.



Huang, M., Carmichael, G. R., Pierce, R. B., Jo, D. S., Park, R. J., Flemming, J., Emmons, L. K.,

Bowman, K. W., Henze, D. K., Davila, Y., Sudo, K., Jonson, J. E., Tronstad Lund, M., Janssens-

Maenhout, G., Dentener, F. J., Keating, T. J., Oetjen, H., and Payne, V. H.: Impact of intercontinental

pollution transport on North American ozone air pollution: an HTAP phase 2 multi-model study,

Atmos. Chem. Phys., 17, 5721-5750, https://doi.org/10.5194/acp-17-5721-2017, 2017.

Jaeglé, L., Wood, R., and Wargan, K.: Multiyear Composite View of Ozone Enhancements and

Stratosphere-to-Troposphere Transport in Dry Intrusions of Northern Hemisphere Extratropical

Cyclones, J. Geophys. Res., 122, 13,436-413,457, https://doi.org/10.1002/2017JD027656, 2017.

Jhun, J.-G., and Lee, E.-J.: A new East Asian winter monsoon index and associated characteristics of the

winter monsoon, J. Climate, 17, 711-726, https://doi.org/10.1175/1520-

0442(2004)017<0711:ANEAWM>2.0.CO;2, 2004.

Jiang, Z., Miyazaki, K., Worden, J. R., Liu, J. J., Jones, D. B. A., and Henze, D. K.: Impacts of

anthropogenic and natural sources on free tropospheric ozone over the Middle East, Atmos. Chem.

Phys., 16, 6537-6546, https://doi.org/10.5194/acp-16-6537-2016, 2016.

Jiang, Z., McDonald, B. C., Worden, H., Worden, J. R., Miyazaki, K., Qu, Z., Henze, D. K., Jones, D. B.

A., Arellano, A. F., Fischer, E. V., Zhu, L., and Boersma, K. F.: Unexpected slowdown of US

pollutant emission reduction in the past decade, Proc. Natl. Acad. Sci. USA, 115, 5099,

https://doi.org/10.1073/pnas.1801191115, 2018.

Jiang, Z., Worden, J. R., Jones, D. B. A., Lin, J. T., Verstraeten, W. W., and Henze, D. K.: Constraints on

Asian ozone using Aura TES, OMI and Terra MOPITT, Atmos. Chem. Phys., 15, 99-112,

https://doi.org/10.5194/acp-15-99-2015, 2015.

Jonson, J. E., Schulz, M., Emmons, L., Flemming, J., Henze, D., Sudo, K., Tronstad Lund, M., Lin, M.,

Benedictow, A., Koffi, B., Dentener, F., Keating, T., Kivi, R., and Davila, Y.: The effects of

intercontinental emission sources on European air pollution levels, Atmos. Chem. Phys., 18, 13655-





13672, https://doi.org/10.5194/acp-18-13655-2018, 2018.

Karamchandani, P., Long, Y., Pirovano, G., Balzarini, A., and Yarwood, G.: Source-sector contributions
to European ozone and fine PM in 2010 using AQMEII modeling data, Atmos. Chem. Phys., 17,
5643-5664, https://doi.org/10.5194/acp-17-5643-2017, 2017.

Kim, M. J., Park, R. J., Ho, C.-H., Woo, J.-H., Choi, K.-C., Song, C.-K., and Lee, J.-B.: Future ozone
and oxidants change under the RCP scenarios, Atmos. Environ., 101, 103-115,

https://doi.org/10.1016/j.atmosenv.2014.11.016, 2015.

Knowland, K. E., Doherty, R. M., Hodges, K. I., and Ott, L. E.: The influence of mid-latitude cyclones
on European background surface ozone, Atmos. Chem. Phys., 17, 12421-12447,
https://doi.org/10.5194/acp-17-12421-2017, 2017.

Lelieveld, J., and Dentener, F. J.: What controls tropospheric ozone?, J. Geophys. Res., 105, 3531-3551,

https://doi.org/10.1029/1999JD901011, 2000.

Li, B., Gasser, T., Ciais, P., Piao, S., Tao, S., Balkanski, Y., Hauglustaine, D., Boisier, J.-P., Chen, Z.,
Huang, M., Li, L. Z., Li, Y., Liu, H., Liu, J., Peng, S., Shen, Z., Sun, Z., Wang, R., Wang, T., Yin, G.,
Yin, Y., Zeng, H., Zeng, Z., and Zhou, F.: The contribution of China's emissions to global climate
forcing, Nature, 531, 357-361, https://doi.org/10.1038/nature17165, 2016.

Li, J., and Zeng, Q.: A unified monsoon index, Geophys. Res. Lett., 29, 115-111-115-114,
https://doi.org/10.1029/2001GL013874, 2002.

Li, J., Yang, W., Wang, Z., Chen, H., Hu, B., Li, J., Sun, Y., Fu, P., and Zhang, Y.: Modeling study of
surface ozone source-receptor relationships in East Asia, Atmos. Res., 167, 77-88,
https://doi.org/10.1016/j.atmosres.2015.07.010, 2016.

Li, Q., Zhang, R., and Wang, Y.: Interannual variation of the wintertime fog–haze days across central
and eastern China and its relation with East Asian winter monsoon, Int. J. Climatol., 36, 346-354,
https://doi.org/10.1002/joc.4350, 2015.



Li, S., Wang, T., Huang, X., Pu, X., Li, M., Chen, P., Yang, X.-Q., and Wang, M.: Impact of East Asian summer monsoon on surface ozone pattern in China, J. Geophys. Res., 123, 1401-1411, https://doi.org/10.1002/2017JD027190, 2018.

Li, X., Liu, J., Mauzerall, D. L., Emmons, L. K., Walters, S., Horowitz, L. W., and Tao, S.: Effects of trans-Eurasian transport of air pollutants on surface ozone concentrations over Western China, J. Geophys. Res., 119, 12,338-312,354, https://doi.org/10.1002/2014JD021936, 2014.

Li, Z., Lau, W. K. M., Ramanathan, V., Wu, G., Ding, Y., Manoj, M. G., Liu, J., Qian, Y., Li, J., Zhou, T., Fan, J., Rosenfeld, D., Ming, Y., Wang, Y., Huang, J., Wang, B., Xu, X., Lee, S. S., Cribb, M., Zhang, F., Yang, X., Zhao, C., Takemura, T., Wang, K., Xia, X., Yin, Y., Zhang, H., Guo, J., Zhai, P. M., Sugimoto, N., Babu, S. S., and Brasseur, G. P.: Aerosol and monsoon climate interactions over Asia, Rev. Geophys., 54, 866-929, https://doi.org/10.1002/2015RG000500, 2016.

Liang, C. K., West, J. J., Silva, R. A., Bian, H., Chin, M., Davila, Y., Dentener, F. J., Emmons, L., Flemming, J., Folberth, G., Henze, D., Im, U., Jonson, J. E., Keating, T. J., Kucsera, T., Lenzen, A., Lin, M., Lund, M. T., Pan, X., Park, R. J., Pierce, R. B., Sekiya, T., Sudo, K., and Takemura, T.: HTAP2 multi-model estimates of premature human mortality due to intercontinental transport of air pollution and emission sectors, Atmos. Chem. Phys., 18, 10497-10520, https://doi.org/10.5194/acp-18-10497-2018, 2018.

Lin, J., Pan, D., Davis, S. J., Zhang, Q., He, K., Wang, C., Streets, D. G., Wuebbles, D. J., and Guan, D.: China's international trade and air pollution in the United States, Proc. Natl. Acad. Sci. USA, 111, 1736-1741, https://doi.org/10.1073/pnas.1312860111, 2014.

Liu, H., Jacob, D. J., Chan, L. Y., Oltmans, S. J., Bey, I., Yantosca, R. M., Harris, J. M., Duncan, B. N., and Martin, R. V.: Sources of tropospheric ozone along the Asian Pacific Rim: An analysis of ozonesonde observations, J. Geophys. Res., 107, D21, https://doi.org/10.1029/2001JD002005, 2002.

Liu, J., Mauzerall, D. L., and Horowitz, L. W.: Analysis of seasonal and interannual variability in



transpacific transport, J. Geophys. Res., 110, D04302, https://doi.org/10.1029/2004JD005207, 2005.

Liu, J. J., Jones, D. B. A., Worden, J. R., Noone, D., Parrington, M., and Kar, J.: Analysis of the summertime buildup of tropospheric ozone abundances over the Middle East and North Africa as

observed by the Tropospheric Emission Spectrometer instrument, J. Geophys. Res., 114, 730-734, https://doi.org/10.1029/2008JD010993, 2009.

Liu, J. J., Jones, D. B. A., Zhang, S., and Kar, J.: Influence of interannual variations in transport on summertime abundances of ozone over the Middle East, J. Geophys. Res., 116, https://doi.org/10.1029/2011JD016188, 2011.

Lu, X., Hong, J., Zhang, L., Cooper, O. R., Schultz, M. G., Xu, X., Wang, T., Gao, M., Zhao, Y., and Zhang, Y.: Severe surface ozone pollution in China: A global perspective, Environ. Sci. Tech. Let., 5, 487-494, https://doi.org/10.1021/acs.estlett.8b00366, 2018.

McLinden, C. A., Olsen, S. C., Hannegan, B., Wild, O., Prather, M. J., and Sundet, J.: Stratospheric ozone in 3-D models: A simple chemistry and the cross-tropopause flux, J. Geophys. Res., 105,

14653-14665, https://doi.org/10.1029/2000JD900124, 2000.

Monks, P. S., Archibald, A. T., Colette, A., Cooper, O., Coyle, M., Derwent, R., Fowler, D., Granier, C., Law, K. S., Mills, G. E., Stevenson, D. S., Tarasova, O., Thouret, V., von Schneidemesser, E., Sommariva, R., Wild, O., and Williams, M. L.: Tropospheric ozone and its precursors from the urban to the global scale from air quality to short-lived climate forcer, Atmos. Chem. Phys., 15, 8889-8973,

https://doi.org/10.5194/acp-15-8889-2015, 2015.

Myhre, G., Aas, W., Cherian, R., Collins, W., Faluvegi, G., Flanner, M., Forster, P., Hodnebrog, Ø., Klimont, Z., Lund, M. T., Mülmenstädt, J., Lund Myhre, C., Olivié, D., Prather, M., Quaas, J., Samset, B. H., Schnell, J. L., Schulz, M., Shindell, D., Skeie, R. B., Takemura, T., and Tsyro, S.: Multi-model simulations of aerosol and ozone radiative forcing due to anthropogenic emission

changes during the period 1990–2015, Atmos. Chem. Phys., 17, 2709-2720,



https://doi.org/10.5194/10.5194/acp-17-2709-2017, 2017.

Nagashima, T., Ohara, T., Sudo, K., and Akimoto, H.: The relative importance of various source regions on East Asian surface ozone, Atmos. Chem. Phys., 10, 11305-11322, https://doi.org/10.5194/acp-10-11305-2010, 2010.

745 Nagashima, T., Sudo, K., Akimoto, H., Kurokawa, J., and Ohara, T.: Long-term change in the source contribution to surface ozone over Japan, Atmos. Chem. Phys., 17, 8231-8246, https://doi.org/10.5194/acp-17-8231-2017, 2017.

Nassar, R., Logan, J. A., Worden, H. M., Megretskaia, I. A., Bowman, K. W., Osterman, G. B., Thompson, A. M., Tarasick, D. W., Austin, S., Claude, H., Dubey, M. K., Hocking, W. K., Johnson,

750 B. J., Joseph, E., Merrill, J., Morris, G. A., Newchurch, M., Oltmans, S. J., Posny, F., Schmidlin, F. J., Vömel, H., Whiteman, D. N., and Witte, J. C.: Validation of Tropospheric Emission Spectrometer (TES) nadir ozone profiles using ozonesonde measurements, J. Geophys. Res., 113, D15S17, https://doi.org/10.1029/2007JD008819, 2008.

Ni, R., Lin, J., Yan, Y., and Lin, W.: Foreign and domestic contributions to springtime ozone over China,

755 Atmos. Chem. Phys., 18, 11447-11469, https://doi.org/10.5194/acp-18-11447-2018, 2018.

Nopmongcol, U., Liu, Z., Stoeckenius, T., and Yarwood, G.: Modeling intercontinental transport of ozone in North America with CAMx for the Air Quality Model Evaluation International Initiative (AQMEII) Phase 3, Atmos. Chem. Phys., 17, 9931-9943, https://doi.org/10.5194/acp-17-9931-2017, 2017.

760 Olivier, J. G. J. and Berdowski, J. J. M.: Global emissions sources and sinks, in: The Climate System, edited by: Berdowski, J., Guicherit, R., and Heij, B.J., 33-78. A. A. Balkema Publishers/Swets & Zeitlinger Publishers, Lisse, the Netherlands., 2001.

Pulles, T, et al., Assessment of Global Emissions from Fuel Combustion in the Final Decades of the 20th Century, TNO report A-R0132/B, Ned. Org.voor toegepast Natuurwet, Onderzoek, Apeldoorn,




the Netherlands, 2007.

Sekiya, T., and Sudo, K.: Role of meteorological variability in global tropospheric ozone during 1970–
    2008, J. Geophys. Res., 117, https://doi.org/10.1029/2012JD018054, 2012.

Sekiya, T., and Sudo, K.: Roles of transport and chemistry processes in global ozone change on
    interannual and multidecadal time scales, J. Geophys. Res., 119, 4903-4921,
https://doi.org/10.1002/2013JD020838, 2014.

Sudo, K. and Akimoto, H.: Global source attribution of tropospheric ozone: Long-range transport from
    various source regions, J. Geophys. Res., 112, https://doi.org/10.1029/2006JD007992, 2007.

Sun, B. and Li, C.: Relationship between the disturbances of East Asian trough and tropical convective
    activities in boreal winter. Chin. Sci. Bull. 42: 500–504 (in Chinese), 1997.

Verstraeten, W. W., Boersma, K. F., Zörner, J., Allaart, M. A. F., Bowman, K. W., and Worden, J. R.:
    Validation of six years of TES tropospheric ozone retrievals with ozonesonde measurements:
    implications for spatial patterns and temporal stability in the bias, Atmos. Meas. Tech., 6, 1413-1423,
    https://doi.org/10.5194/amt-6-1413-2013, 2013.

Verstraeten, W. W., Neu, J. L., Williams, J. E., Bowman, K. W., Worden, J. R., and Boersma, K. F.:
Rapid increases in tropospheric ozone production and export from China, Nat. Geosci., 8, 690-695,
    https://doi.org/10.1038/ngeo2493, 2015.

Vogel, B., Günther, G., Müller, R., Grooß, J. U., and Riese, M.: Impact of different Asian source regions
    on the composition of the Asian monsoon anticyclone and of the extratropical lowermost
    stratosphere, Atmos. Chem. Phys., 15, 13699-13716, https://doi.org/10.5194/acp-15-13699-2015,
785     2015.

Wang, B. and Fan, Z.: Choice of South Asian Summer Monsoon Indices, B. Am. Meteorol. Soc., 80,
    629-638, https://doi.org/10.1175/1520-0477(1999)080<0629:COSASM>2.0.CO;2, 1999.

Wang, B., Wu, Z., Li, J., Liu, J., Chang, C.-P., Ding, Y., and Wu, G.: How to Measure the Strength of the



East Asian Summer Monsoon, J. Climate, 21, 4449-4463, https://doi.org/10.1175/2008JCLI2183.1,
790    2008.

Wang, H. and Jiang, D.: A new East Asian winter monsoon intensity index and atmospheric circulation
    comparison between strong and weak composite. Quat. Sci. 24: 19-27 (in Chinese), 2004.

Wang, Y., Zhang, Y., Hao, J., and Luo, M.: Seasonal and spatial variability of surface ozone over China:
    contributions from background and domestic pollution, Atmos. Chem. Phys., 11, 3511-3525,
https://doi.org/10.5194/acp-11-3511-2011, 2011.

Wild, O., Pochanart, P., and Akimoto, H.: Trans-Eurasian transport of ozone and its precursors, J.
    Geophys. Res., 109, D11302, https://doi.org/10.1029/2003JD004501, 2004.

Worden, H. M., Bowman, K. W., Worden, J. R., Eldering, A., and Beer, R.: Satellite measurements of
    the clear-sky greenhouse effect from tropospheric ozone, Nat. Geosci., 1, 305-308,
https://doi.org/10.1038/ngeo182, 2008.

Yang, Y., Liao, H., and Li, J.: Impacts of the East Asian summer monsoon on interannual variations of
    summertime surface-layer ozone concentrations over China, Atmos. Chem. Phys., 14, 6867-6879,
    https://doi.org/10.5194/acp-14-6867-2014, 2014.

Yoshitomi, M., Wild, O., and Akimoto, H.: Contributions of regional and intercontinental transport to
surface ozone in the Tokyo area, Atmos. Chem. Phys., 11, 7583-7599, https://doi.org/10.5194/acp-11-
    7583-2011, 2011.

Zhang, L., Jacob, D. J., Boersma, K. F., Jaffe, D. A., Olson, J. R., Bowman, K. W., Worden, J. R.,
    Thompson, A. M., Avery, M. A., Cohen, R. C., Dibb, J. E., Flock, F. M., Fuelberg, H. E., Huey, L. G.,
    McMillan, W. W., Singh, H. B., and Weinheimer, A. J.: Transpacific transport of ozone pollution and
the effect of recent Asian emission increases on air quality in North America: an integrated analysis
    using satellite, aircraft, ozonesonde, and surface observations, Atmos. Chem. Phys., 8, 6117-6136,
    https://doi.org/10.5194/acp-8-6117-2008, 2008.



Zhang, Q., Jiang, X., Tong, D., Davis, S. J., Zhao, H., Geng, G., Feng, T., Zheng, B., Lu, Z., Streets, D. G., Ni, R., Brauer, M., van Donkelaar, A., Martin, R. V., Huo, H., Liu, Z., Pan, D., Kan, H., Yan, Y., Lin, J., He, K., and Guan, D.: Transboundary health impacts of transported global air pollution and international trade, Nature, 543, 705-709, https://doi.org/10.1038/nature21712, 2017.

Zhang, Q., Streets, D. G., Carmichael, G. R., He, K. B., Huo, H., Kannari, A., Klimont, Z., Park, I. S., Reddy, S., Fu, J. S., Chen, D., Duan, L., Lei, Y., Wang, L. T., and Yao, Z. L.: Asian emissions in 2006 for the NASA INTEX-B mission, Atmos. Chem. Phys., 9, 5131-5153, https://doi.org/10.5194/acp-9-5131-2009, 2009.

Zhang, Q. Y., Tao, S. Y., and Chen, L. T.: The inter-annual variability of East Asian summer monsoon indices and its association with the pattern of general circulation over East Asia (in Chinese). Acta Meteorol. Sin., 61, 559-568, https://doi.org/10.11676/qxxb2003.056, 2003.

Zhang, Y., Kuang, X., Guo, W., and Zhou, T.: Seasonal evolution of the upper-tropospheric westerly jet core over East Asia, Geophys. Res. Lett., 33, L11708, https://doi.org/10.1029/2006GL026377, 2006.

Zheng, B., Tong, D., Li, M., Liu, F., Hong, C., Geng, G., Li, H., Li, X., Peng, L., Qi, J., Yan, L., Zhang, Y., Zhao, H., Zheng, Y., He, K., and Zhang, Q.: Trends in China's anthropogenic emissions since 2010 as the consequence of clean air actions, Atmos. Chem. Phys., 18, 14095-14111, https://doi.org/10.5194/acp-18-14095-2018, 2018.

Zhou, D., Ding, A., Mao, H., Fu, C., Wang, T., Chan, L. Y., Ding, K., Zhang, Y., Liu, J., Lu, A., and Hao, N.: Impacts of the East Asian monsoon on lower tropospheric ozone over coastal South China, Environ. Res. Lett., 8, 044011, https://doi.org/10.1088/1748-9326/8/4/044011, 2013.

Zhu, B., Hou, X., and Kang, H.: Analysis of the seasonal ozone budget and the impact of the summer monsoon on the northeastern Qinghai-Tibetan Plateau, J. Geophys. Res., 121, 2029-2042, https://doi.org/10.1002/2015JD023857, 2016.

Zhu, J., Liao, H., Mao, Y., Yang, Y., and Jiang, H.: Interannual variation, decadal trend, and future



change in ozone outflow from East Asia, Atmos. Chem. Phys., 17, 3729-3747,

https://doi.org/10.5194/acp-17-3729-2017, 2017.

Zhu, Y., Liu, J., Wang, T., Zhuang, B., Han, H., Wang, H., Chang, Y., and Ding, K.: The Impacts of

Meteorology on the Seasonal and Interannual Variabilities of Ozone Transport From North America

to East Asia, J. Geophys. Res., 122, 10,612-610,636, https://doi.org/10.1002/2017JD026761, 2017.



Table 1. GEOS-Chem simulations and experiments in this study[1].

| Simulation Type | Period | Experiment | Description |
| --- | --- | --- | --- |
| Full chemistry simulation | 2005 (2004 for spin-up) | 1. CTRL | Including all emissions |
| | | 2. EAnth-GLO | Excluding global anthropogenic emissions |
| | | 3. EAnth-EAS | Excluding anthropogenic emissions in EAS |
| | | 4. EAnth-NAM | Excluding anthropogenic emissions in NAM |
| | | 5. EAnth-EUR | Excluding anthropogenic emissions in EUR |
| | | 6. EAnth-AFR | Excluding anthropogenic emissions in AFR |
| | | 7. EAnth-CAS | Excluding anthropogenic emissions in CAS |
| | | 8. EAnth-SAS | Excluding anthropogenic emissions in SAS |
| | | 9. EAnth-SEAS | Excluding anthropogenic emissions in SEAS |
| | | 10. EAnth-ROW | Excluding anthropogenic emissions in ROW |
| Tagged ozone simulation | 1987-2006 (1986 for spin-up) | 11. Fix-Chem | Fixing ozone production and loss data in 2005 |

[1]The abbreviations stand for different regions, including East Asia (EAS), North America (NAM),

Europe (EUR), Africa (AFR), central Asia (CAS), South Asia (SAS), Southeast Asia (SEAS), and the

rest of the world (ROW) (see Figure 1). "CTRL" represents "a controlled run". "EAnth" means

"Excluding anthropogenic emissions". In "Fix-Chem" simulations, there is no year-to-year variation in

chemistry.




Table 2. Tropospheric ozone and its components by source and associated terms defined in this paper. The components indicated in italics are the focus in this study.

| | *Tropospheric ozone[1]* | | | |
|---|---|---|---|---|
| By in/out of East Asia | *Foreign ozone[2]* | | *Native ozone[3]* | Stratospheric ozone[4] |
| By foreign region | *A foreign region's name + ozone[5]* | | Native ozone | Stratospheric ozone |
| By precursor | *Anthropogenic ozone[6]* | | Natural ozone[7] | |
| By both precursor and in/out of East Asia | *Foreign anthropogenic ozone[8]* | *Native anthropogenic ozone[9]* | Natural ozone | |
| By both precursor and foreign region | *A foreign region's name +anthropogenic ozone[10]* | Native anthropogenic ozone | Natural ozone | |

[1]Tropospheric ozone refers to ozone in the troposphere and is also termed ozone in this paper, unless

stated otherwise. It is the sum of the ozone components in each row.

[2]Foreign ozone: ozone produced in the troposphere outside East Asia.

[3]Native ozone: ozone produced in the troposphere inside East Asia.

[4]Stratospheric ozone: ozone produced in the stratosphere and then transported to the troposphere.

[5]A foreign region's name + ozone: ozone produced in the troposphere over that foreign region. For

example, European ozone.

[6]Anthropogenic ozone: ozone produced from anthropogenic precursors.



[7]Natural ozone: ozone produced from natural precursors and from the stratosphere.

[8]Foreign anthropogenic ozone: ozone produced from anthropogenic precursors emitted outside East Asia.

[9]Native anthropogenic ozone: ozone produced from anthropogenic precursors emitted inside East Asia.

[10]A foreign region's name + anthropogenic ozone: ozone produced from anthropogenic precursors emitted from that foreign region. For example, European anthropogenic ozone.




Table 3. The contributions of foreign ozone from different regions to surface ozone in East Asia in 2005. The fractional contributions are indicated in brackets, taking surface ozone in East Asia as 100%[1].

| | | Annual | Spring | Summer | Autumn | Winter |
|---|---|---|---|---|---|---|
| Native ozone | | 20.4 (46.4%) | 20.7 (43.1%) | 30.1 (64.2%) | 19.2 (49.1%) | 11.5 (28.1%) |
| Foreign ozone | Total | 22.2 (50.5%) | 25.4 (52.9%) | 16.5 (35.3%) | 19.7 (50.4%) | 27.1 (66.4%) |
| | NAM | 3.4 (7.8%) | 3.7 (7.7%) | 1.2 (2.6%) | 3.3 (8.4%) | 5.7 (14.0%) |
| | EUR | 4.3 (9.7%) | 5.1 (10.6%) | 2.3 (4.9%) | 4.2 (10.8%) | 5.3 (13.0%) |
| | AFR | 1.2 (2.8%) | 1.2 (2.6%) | 0.3 (0.7%) | 0.8 (2.1%) | 2.4 (5.9%) |
| | CAS | 4.0 (9.1%) | 4.1 (8.5%) | 4.7 (10.1%) | 3.8 (9.8%) | 3.3 (8.0%) |
| | SAS | 2.1 (4.8%) | 2.9 (6.1%) | 2.0 (4.2%) | 1.6 (4.0%) | 2.2 (5.5%) |
| | SEAS | 0.9 (2.0%) | 1.0 (2.1%) | 1.3 (2.8%) | 0.7 (1.7%) | 0.7 (1.7%) |
| | ROW | 6.2 (14.2%) | 7.4 (15.4%) | 4.7 (9.9%) | 5.3 (13.7%) | 7.5 (18.3%) |
| Stratospheric ozone | | 1.3 (3.0%) | 2.1 (4.3%) | 0.3 (0.7%) | 0.4 (1.0%) | 2.5 (6.2%) |
| Surface ozone | | 43.9 (100%) | 48.0 (100%) | 46.9 (100%) | 39.1 (100%) | 40.8 (100%) |

[1]All values are the means over East Asia. The abbreviations stand for different regions (Figure 1), including East Asia (EAS), North America (NAM), Europe (EUR), Africa (AFR), central Asia (CAS), South Asia (SAS), Southeast Asia (SEAS), and the rest of the world (ROW).





Table 4. The contributions of anthropogenic ozone (in ppbv) from different regions to surface ozone in East Asia in 2005. The fractional contributions are indicated in brackets, taking surface ozone in East Asia as 100%[1].

|  |  | Annual | Spring | Summer | Autumn | Winter |
|---|---|---|---|---|---|---|
| Native anthropogenic ozone |  | 6.3 (15.2%) | 6.9 (15.1%) | 9.8 (21.8%) | 6.2 (16.6%) | 2.2 (5.9%) |
| Foreign anthropogenic ozone | Total | 4.7 (11.3%) | 6.4 (14.1%) | 3.7 (8.2%) | 4.1 (10.8%) | 4.6 (12.1%) |
|  | NAM | 1.3 (3.1%) | 1.7 (3.7%) | 0.6 (1.4%) | 1.1 (2.9%) | 1.8 (4.6%) |
|  | EUR | 1.0 (2.4%) | 1.8 (3.9%) | 0.7 (1.6%) | 1.0 (2.6%) | 0.6 (1.6%) |
|  | AFR | 0.2 (0.4%) | 0.2 (0.4%) | 0.2 (0.4%) | 0.1 (0.4%) | 0.2 (0.5%) |
|  | CAS | 0.6 (1.5%) | 1.0 (2.2%) | 0.5 (1.2%) | 0.7 (1.8%) | 0.3 (0.9%) |
|  | SAS | 0.7 (1.8%) | 0.9 (2.0%) | 0.7 (1.6%) | 0.5 (1.4%) | 0.8 (2.1%) |
|  | SEAS | 0.5 (1.3%) | 0.5 (1.1%) | 0.7 (1.6%) | 0.5 (1.2%) | 0.5 (1.3%) |
|  | ROW | 0.3 (0.8%) | 0.4 (0.8%) | 0.2 (0.5%) | 0.2 (0.6%) | 0.4 (1.2%) |
| Natural ozone |  | 30.5 (73.6%) | 32.0 (70.8%) | 31.4 (70.0%) | 27.2 (72.5%) | 31.2 (82.0%) |
| Surface ozone |  | 41.5 (100%) | 45.4 (100%) | 44.9 (100%) | 37.5 (100%) | 38.1 (100%) |

[1]All values are the means over East Asia. The abbreviations stand for different regions (Figure 1), including East Asia (EAS), North America (NAM), Europe (EUR), Africa (AFR), central Asia (CAS), South Asia (SAS), Southeast Asia (SEAS), and the rest of the world (ROW).




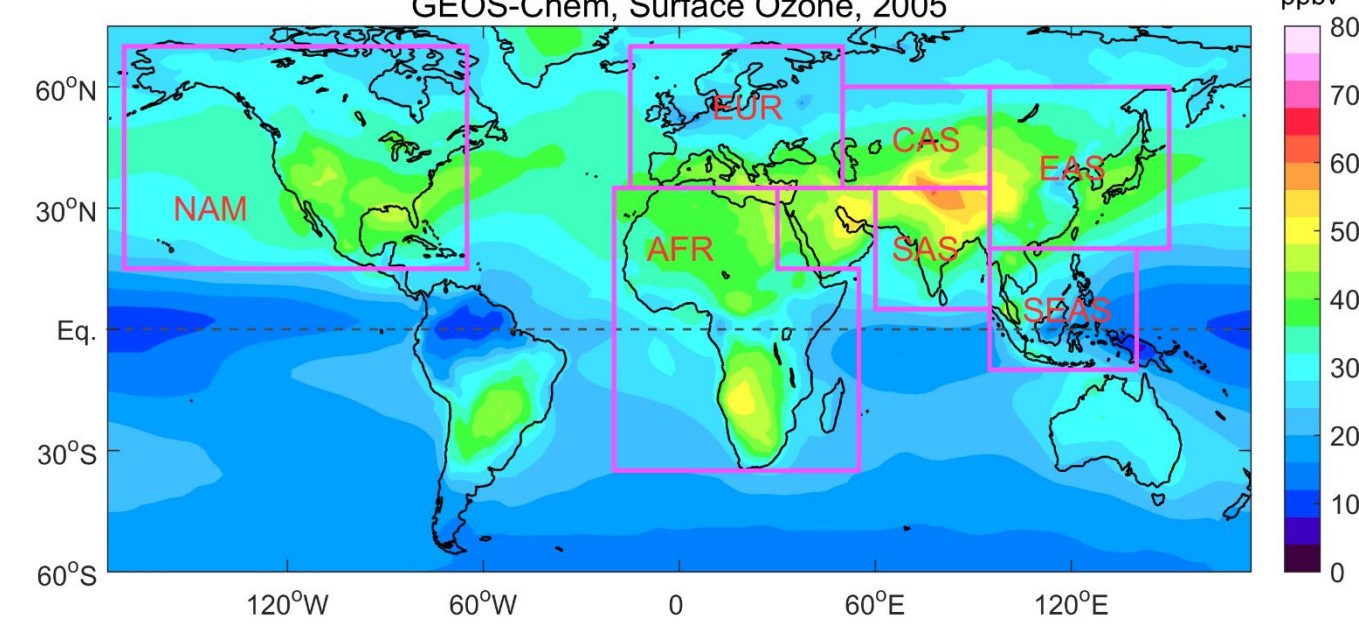

Figure 1. Annual mean surface ozone in 2005 from GEOS-Chem simulations. The purple boxed areas
define seven regions, plus the rest of the world (ROW, all regions except the boxed regions). The seven
regions include East Asia (EAS, 95°E-150°E, 20°N-60°N), North America (NAM, 170°W-65°W, 15°N-
70°N), Europe (EUR, 15°E-50°E, 35°N-70°N), Africa (AFR, 20°W-30°E, 0-35°N and 20°W-55°E,
35°S-0), central Asia (CAS, 50°E-95°E, 35°N-60°N), South Asia (SAS, 60°E-95°E, 5°N-35°N), and
Southeast Asia (SEAS, 95°E-140°E, 10°S-20°N).








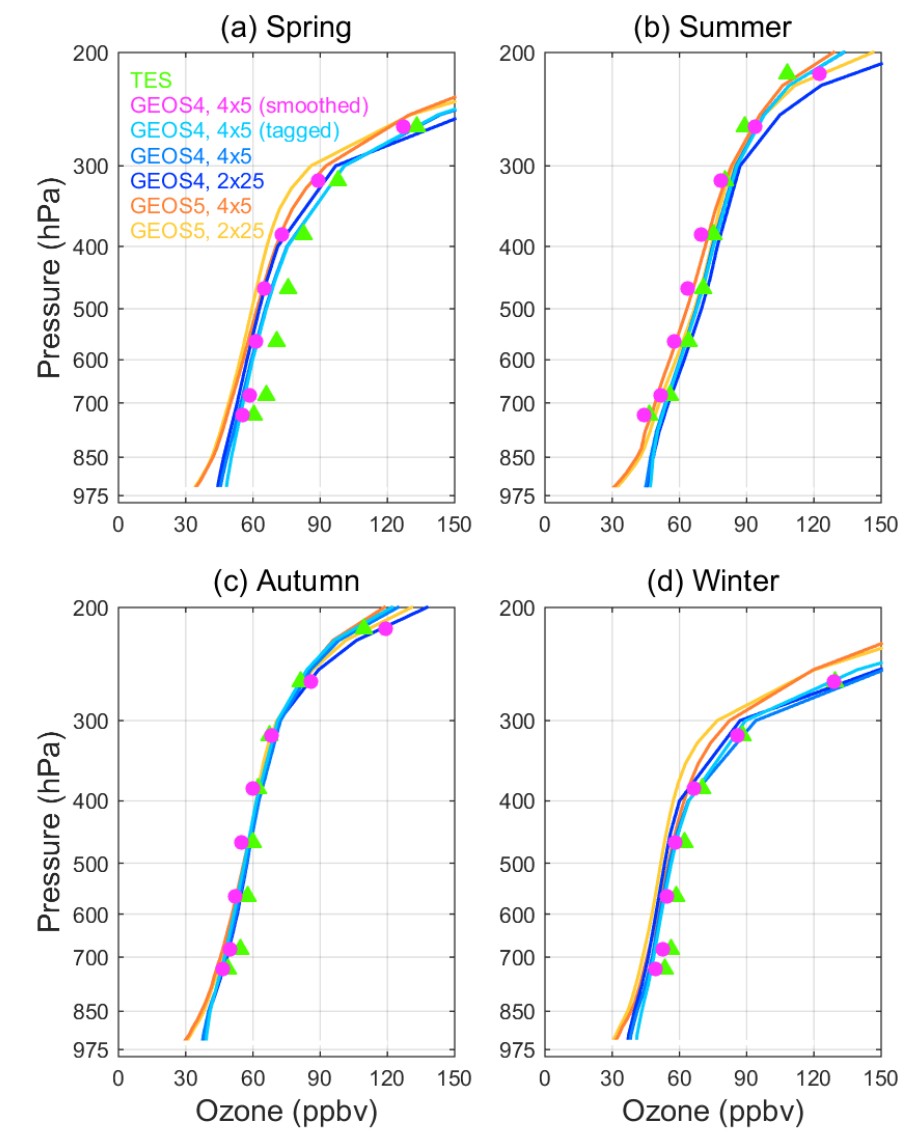

Figure 2. Vertical ozone profiles averaged over East Asia in 2005 from GEOS-Chem simulations driven by GEOS-4 and GEOS-5 meteorology data with different horizontal resolutions (4° by 5° and 2° by 2.5° in latitude and longitude, indicated in different colours). TES retrievals (in green) and GEOS-Chem simulations smoothed with the TES *a priori* and averaging kernels (in purple) are also shown.









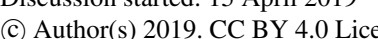

Figure 3. (a-d) Mean vertical profiles of native ozone (in blue) and foreign ozone (in red) (in ppbv) over East Asia by season in 2005. (e-h) The same as (a-d), but for the fractional contribution (in %) of each

component to ozone in the East Asian troposphere at each altitude.


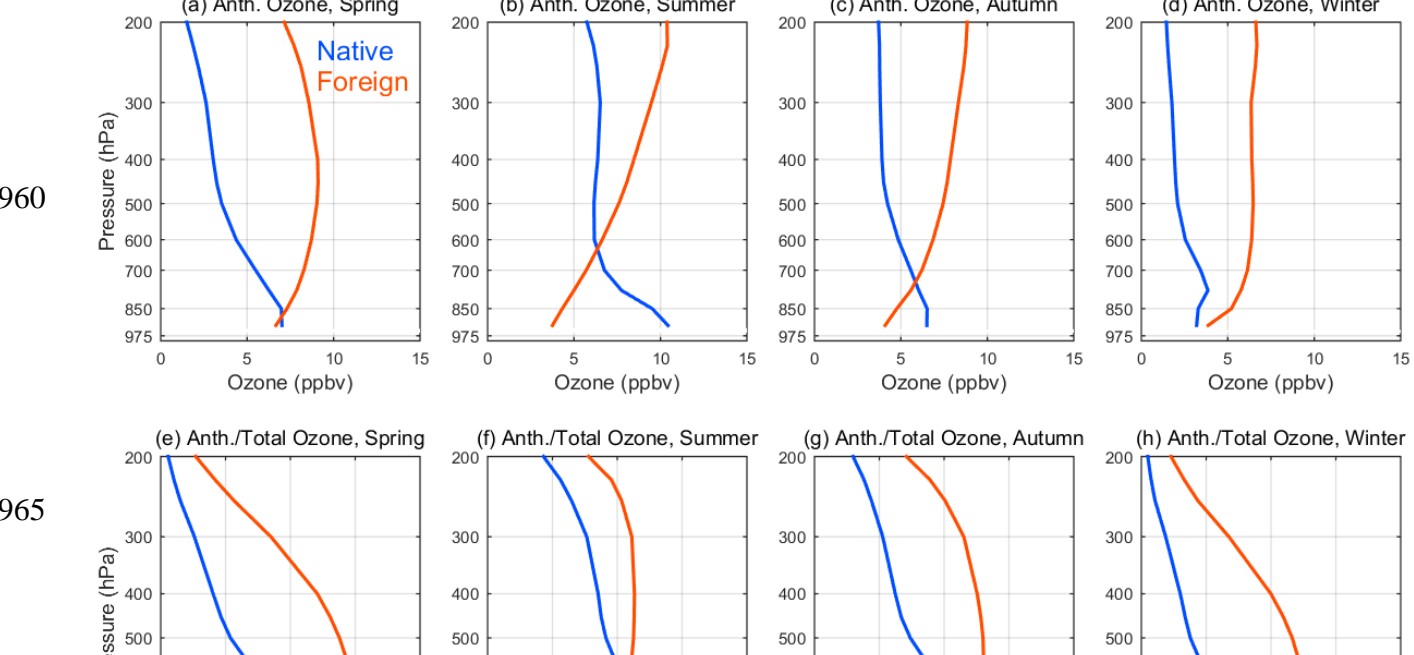

Figure 4. (a-d) Mean vertical profiles of native anthropogenic ozone (in blue) and foreign anthropogenic ozone (in red) (in ppbv) over East Asia by season in 2005. (e-h) The same as (a-d), but for the fractional contribution (in %) of each component to ozone in the East Asian troposphere at each altitude. Note the difference in the scale of the x-axis between Figure 3 and Figure 4.





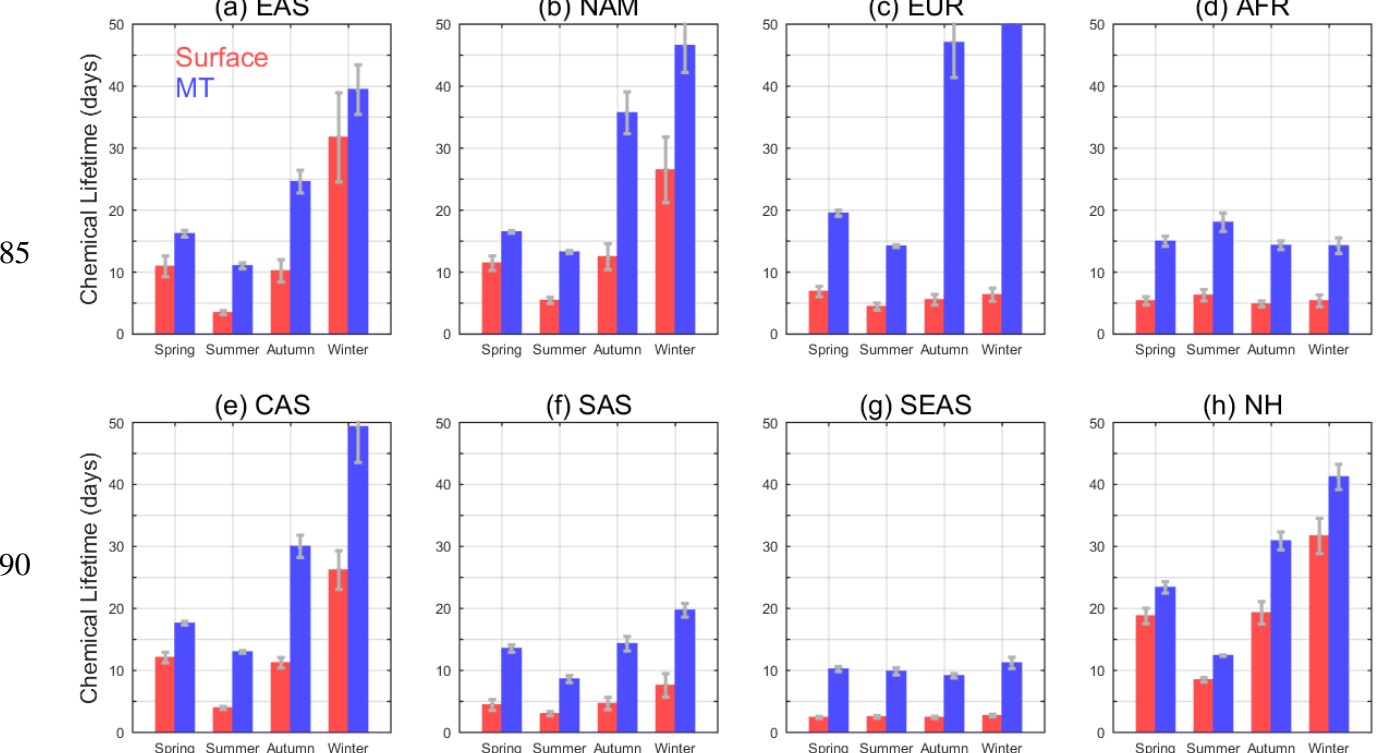

Figure 5. Chemical lifetimes of ozone averaged in various regions at the surface and in the middle

troposphere (MT, 500 hPa) in 2005, calculated from the ozone loss rate in GEOS-Chem simulations.

The error bar is 4 times the standard error of the mean (equivalent to the 95% confidence limit of the

mean). The regions include East Asia (EAS), North America (NAM), Europe (EUR), Africa (AFR),

central Asia (CAS), South Asia (SAS), Southeast Asia (SEAS), and the Northern Hemisphere (NH).








Figure 6. Longitude-pressure cross-sections of foreign ozone (in ppbv, in colour) and wind field (in arrows) averaged between 20-40°N (a-d) and 40-60°N (e-h) in 2005. The white areas indicate topography. The blue lines indicate the western border of East Asia. The vertical velocities in the p

coordinates are enlarged 1000 times for illustration purposes.





Figure 7. (a-d) Vertical variations of ozone (in ppbv) over East Asia from native and foreign sources during the four seasons. (e-h) The same as (a-d), but in terms of the factional contributions (in %) to ozone produced in the troposphere over East Asia. (i-l) The same as (a-d) but in terms of the fractional contributions (in %) to ozone over East Asia. All values are the means over East Asia in 2005. The abbreviations are for East Asia (EAS), North America (NAM), Europe (EUR), Africa (AFR), central Asia (CAS), South Asia (SAS), Southeast Asia (SEAS), the rest of the world (ROW), and the stratosphere (STR).





Figure 8. The fractional contributions (in %, left y-axis) of native and foreign sources to ozone in East Asia at the surface (a-d) and in the middle troposphere (MT, 500 hPa) (e-h) in 2005. The black lines indicate ozone concentrations averaged over East Asia (in ppbv, right y-axis). The abbreviations are for East Asia (EAS), North America (NAM), Europe (EUR), Africa (AFR), central Asia (CAS), South Asia (SAS), Southeast Asia (SEAS), the rest of the world (ROW), and the stratosphere (STR).









Figure 9. Horizontal distributions of ozone produced in different regions overlaid with streamlines in the upper troposphere (UT, 200 hPa) in the summer of 2005. The boxed area indicates East Asia. The abbreviations are for East Asia (EAS), North America (NAM), Europe (EUR), Africa (AFR), central Asia (CAS), South Asia (SAS), Southeast Asia (SEAS), and the rest of the world (ROW).



Figure 10. Interannual variations of the anomalies of foreign ozone (in ppbv, left y-axis) from various regions at the East Asian surface driven by meteorology (Fix-Chem) in winter (a) and summer (b). The values are averaged over 110-145°E, 22-46°N in winter and 95-130°E, 22-46°N in summer, where the correlations are the most significant. The correlation coefficients are calculated with three sets of monsoon indices, separately, in winter and summer. The abbreviations are for East Asia (EAS), North America (NAM), Europe (EUR), South Asia (SAS), and Southeast Asia (SEAS).