# Peer review of "Foreign influences on tropospheric ozone over East Asia through global atmospheric transport"

_Atmospheric Chemistry and Physics, 2019_

## Referee Comment (RC1) · Anonymous Referee #2 · 13 May 2019

This publication represents a robust analysis of contributions to surface and tropospheric ozone. While several prior studies have found similar results, and in this sense the study is maybe not completely new, the comprehensive analysis of contributors to ozone over Asia has clearly added value.

On the downside, I think it is regrettable that the authors have not attempted to align their study better with the HTAP2 source-receptor studies, that included harmonized simulations of emission sensitivities and responses over East Asia and a number of other world regions, updated and harmonized emissions, etc. As a result, it is becoming more difficult to evaluate uncertainties related to the use of one specific model compared to other models.

Nevertheless, I find the overall analysis convincing, the material well presented, and

therefore recommend to publish the manuscript in ACP, with some suggestions for minor revision presented below.

Minor suggestions:

l. 17 East Asia defined as . . ..

l. 48 the ecosystem=>ecosystems.

l. 54 why?

l. 57 I don't find this in Fiore's paper. Anyway as the ozone response depends on the emission reduction strength, the sensivity will depend on the magnitude of perturbation. In this sense perturbations that are close to the present situation (i.e. a 5, 10 or 20 %) are used for Source-receptor relationships, sometimes with a correction for larger perturbation sizes. These have been used e.g. in HTAP1, and HTAP2, and other studies by individual researchers also for Asia. A nice paper that combines source attribution and tagging paper: https://www.geosci-model-dev.net/11/2825/2018/. It can be used to address some of the uncertainties. I also note that it is not very clear around l. 154 how exactly the tagging was done- and there are several ways to do so.

l. 98. It is confusing to talk about trends when you really talk about interannual variability.

l. 127 I notice that this resolution is meanwhile not really state-of-the-art. Mention already here that you also do higher resolution sensitivity simulations

l. 130 Unfortunately the different choice of regions compared to HTAP1 or HTAP2 does not help in comparing to other model simulations.

142-145 I guess what the authors are doing here is introducing a correction factor so that the sum of the individual region 100 % perturbations nicely sums up to a global perturbation? How large are these corrections?

l. 164. Linoz. Does this effectively mean a constant yearly influx of stratospheric ozone

by 484 Tg in these variation. Note that it is likely that there is a correlation between large scale circulation (and I guess monsoon as well) and strat-trop exchange.

l. 137 Why were 2006/2007 chosen? And not more recent years (e.g. HTAP's 2010)

l. 157 No variation in chemistry. In the next line the authors explain this is achieved by extracting production/loss data for 2005. This is going to lead to inconsistencies (and hopefully identical results for 2005). This needs discussion.

l. 170. The results of using different resolutions and meteorological drivers, needs to be somehow included in the discussion of uncertainties.

l. 194 here the operational definition of troposphere needs to be given.

l. 202-204 In this case North American ozone is both from natural and anthropogenic emissions as well as stratospheric ozone that entered the North American region?

l. 330 I recommend also to consider the results of Turnock et al, ACP, 2018, which discusses HTAP2 results.

l. 346. This finding warrant a bit more discussion, given the similarity of emissions but longer distance compared to Europe.

l. 368 I guess that can well be, also given the fact that many other models have larger stratospheric influxes. Would the conclusions change substantially if the number would be double. It is unclear to me how stratospheric ozone influx is accounted in the various attribution methods (e.g. influx of ozone In Europe, is that European ozone as well?)

l. 410 It is not so clear why you would need 3x2 monsoon indices. If they are so different, please summarize what aspects they would represent stronger. Please also clarify how monsoon (summer phenomenon) indices also can have significant correlation with winter ozone.

l. 540 It would have been great to use the HTAP2 compilation for 2008/2010- as did many other models. It would be good to mention at least the differences.

---

## Referee Comment (RC2) · Anonymous Referee #1 · 15 May 2019

General Comment

This paper examined the tropospheric ozone in East Asia in terms of the influence of various source regions, particularly focused on the regions outside East Asia, they called them as "foreign ozone". The topic of the paper is well within the scope of the journal. The model used and the methods to deduce the influence of each different source region are adequate and have been applied so far in various studies for similar purpose. The results shown in the paper are generally consistent with the facts published in the previous literatures. This gives a certain reliability to the analysis done in this paper, but at the same time, the novelty of this paper over those previous studies is not clearly shown in the manuscript. For example, the authors reviewed the roles of East Asian Monsoon on the foreign ozone in East Asia in "Introduction", but the find-

ings of this paper written in "Abstract" is quite similar to what was reviewed there. The author argued that this paper provided a comprehensive assessment of the influence of foreign ozone on the East Asian tropospheric ozone, but if they just want a comprehensive assessment, writing a review paper is more suitable, and actually the review given in this paper is quite comprehensive. I suggest the authors should state more clearly what they achieved on top of the previous studies to be published in the journal. This is my general concern.

Major Comment:

Almost all analysis was done for the average over East Asia. As previous studies have shown that the relative contributions of various source regions can vary considerably depending on the location within East Asia. So, I cannot fully understand the meaning of such "East Asian averaged" contributions of various source regions. Actually, the latitudinal dependence in each foreign and native contributions were analyzed, but I guess the longitudinal dependence should be also large enough to be analyzed.

Specific Comments:

- L45: The terms "native" and "foreign" are used before their definition is given in Table 2.

- L83-85: Is the "transport" itself associated with thing other than meteorology such as emission and/or chemistry?

-L124-125: How did you treat CH4 chemistry in the model? Is it fully represented?

- L132-136: I don't think these detailed definition of regions are necessary, since Figure 1 shows them visually and they can be found in its caption.

- L146 (formula (1)): How is the consistency between CTRL-EAnth-GLO and the sum in the denominator of the first term? How are they close to each other? Should be explained somewhere in the text.

- L159-160: The production and loss of ozone can vary considerably with in a day, so I imagine using daily production and loss data should have bad consequences on the simulated tagged ozone concentration. Did you check the validity for using the daily values for them?

- Table 2: I don't think Table 2 could effectively explain the different definition of the terminology used in the manuscript. For me, the caption of Table 2 is easier to understand what you want to explain than Table2 itself and the descriptions in the main text (L194-210).

- L233-234: You can not compare the contribution of foreign ozone on whole East Asia and that on China

- L248: How did you calculate the chemical lifetime? Explain it in the text.

- L251-252: Only the dry deposition could be the cause of the difference? Are there any other causes which should be mentioned here?

- L267-269: This part should be more specific. Where is the East Asian trough? Which region of downdraft you are referring?

- L284-287: This sentence is quite hard to understand logically. The difference between foreign and foreign anthropogenic O3 is not mentioned in the previous sentences. I cannot understand what you want to mean here.

- L342: Why the contribution from Africa in winter can be so large? Why it can be larger than that from SAS SEAS where much closer to EAS.

- L442 (formula(8)): What is U'850?

- L484-487: Where should I look at in Figure 3 and 7 to find these values? Is it annual evaluation or seasonal? This is a quite blur sentence.

[Figure]

2019.

---

## Author Comment (AC1) · 23 Jul 2019

*We are very grateful to the reviewers for their valuable comments and suggestions, which have helped us greatly in improving our manuscript. We have addressed all the comments and revised the manuscript accordingly. The point-to-point responses are provided below in Italic.*

Anonymous Referee #2

This publication represents a robust analysis of contributions to surface and tropospheric ozone. While several prior studies have found similar results, and in this sense the study is maybe not completely new, the comprehensive analysis of contributors to ozone over Asia has clearly added value.

On the downside, I think it is regrettable that the authors have not attempted to align their study better with the HTAP2 source-receptor studies, that included harmonized simulations of emission sensitivities and responses over East Asia and a number of other world regions, updated and harmonized emissions, etc. As a result, it is becoming more difficult to evaluate uncertainties related to the use of one specific model compared to other models.

Nevertheless, I find the overall analysis convincing, the material well presented, and therefore recommend to publish the manuscript in ACP, with some suggestions for minor revision presented below.

Minor suggestions:

l. 17 East Asia defined as : : :.

*We added the geographic boundaries to define the domain of East Asia in this study.*

l. 48 the ecosystem=>ecosystems.

*Thanks. The sentence has been revised.*

l. 54 why?

*This sentence is revised to explain why.*

l. 57 I don't find this in Fiore's paper. Anyway as the ozone response depends on the emission reduction strength, the sensivity will depend on the magnitude of perturbation. In this sense perturbations that are close to the present situation (i.e. a 5, 10 or 20 %) are used for Source-receptor relationships, sometimes with a correction for larger perturbation sizes. These have been used e.g. in HTAP1, and HTAP2, and other studies by individual researchers also for Asia. A nice paper that combines source attribution and tagging paper: https://www.geosci-model-dev.net/11/2825/2018/. It can be used to address some of the uncertainties. I also note that it is not very clear around

*Thanks for this point. It is better to call this method as "perturbation (sensitivity) method "as suggested by Butler et al. (2018). Reducing the emissions of ozone precursors in source regions to zero in this study is one of the scenarios of the perturbation method. In the manuscript, the 'emission zero-out' has been revised as 'emission perturbation'.*

l. 154 how exactly the tagging was done- and there are several ways to do so.

*In this study, ozone molecules were tagged based on the geographical model domains in which the ozone molecules are formed (Wang et al., 1998). Using the daily ozone production and loss data archived from a full chemical simulation conducted beforehand, net ozone production at each model grid was resolved. For a specific source region A, ozone produced in A is labelled as a tracer. The tracer excludes the ozone molecules formed outside A. Therefore, how the tracer distributes spatially directly show the ozone transport from A to the outside. The amount of the tracer in a receptor region can be directly attributed to ozone production in A. Overall, the tagged ozone simulation in this study tracks ozone produced in the troposphere over each of the defined source regions along its transport into a receptor region. The*

*explanation is added in the last paragraph of section 2.*

l. 98. It is confusing to talk about trends when you really talk about interannual variability.

*Thanks for the point. The discussion about the trend has been deleted.*

l. 127 I notice that this resolution is meanwhile not really state-of-the-art. Mention already here that you also do higher resolution sensitivity simulations

*Thanks. We added statements on our run at higher resolutions.*

l. 130 Unfortunately the different choice of regions compared to HTAP1 or HTAP2 does not help in comparing to other model simulations.

*Definition of the study domains in this study is slightly different to that in HTAP1, but with similarities (Table R1). The comparison has been added.*

*Table R1. Comparison of the definition of the study domains between this study and HTAP1.*

|  | This study | HTAP1 |
|---|---|---|
| East Asia | 95°E-150°E, 20°N-60°N | 95°E-160°E, 15°N-50°N |
| North America | 170°W-65°W, 15°N-70°N | 125°W-60°W, 15°N-55°N |
| Europe | 15°W-50°E, 35°N-70°N | 10°W-50°E, 25°N-65°N |
| South Asia | 60°E-95°E, 5°N-35°N | 50°E-95°E, 5°N-35°N |

142-145 I guess what the authors are doing here is introducing a correction factor so that the sum of the individual region 100 % perturbations nicely sums up to a global perturbation? How large are these corrections?

*Yes. In Equation (1), $\sum_{i=1}^{8}$ (CTRL - EAnth-$X_i$) is the sum of the ozone response to the 100% perturbation at each of the defined regions, CTRL - EAnth-GLO is the ozone response to the 100% perturbation for the globe. In the East Asian troposphere, ozone*

*concentration from $\sum_{i=1}^{8}$ (CTRL - EAnth-$X_i$) is 0-4 ppbv (0-20%) higher than that from CTRL - EAnth-GLO (Figure S1). For each source region, the correction over East Asia is less than 1 ppbv. The magnitude of these corrections are added.*

l. 164. Linoz. Does this effectively mean a constant yearly influx of stratospheric ozone by 484 Tg in these variation. Note that it is likely that there is a correlation between large scale circulation (and I guess monsoon as well) and strat-trop exchange.
*Thanks. The influx of stratospheric ozone to the troposphere varies interannually and is ~484 Tg in 2005. The explanation has been added.*

l. 137 Why were 2006/2007 chosen? And not more recent years (e.g. HTAP's 2010)
*Thanks. In this study, GEOS-Chem was driven by GEOS-4 meteorological data, which has strong performance in simulating tropospheric ozone (Choi et al., 2017; Y. Zhu et al., 2017; Han et al., 2018). GEOS-4 covers 1985-2006, which is the study period here.*

l. 157 No variation in chemistry. In the next line the authors explain this is achieved by extracting production/loss data for 2005. This is going to lead to inconsistencies (and hopefully identical results for 2005). This needs discussion.
*Thanks. Daily ozone production and loss data in 2005 were used for all the years from 1986 to 2006. Therefore, in the simulations, the daily data in 2005 allow a seasonal variation, but no interannual variation in chemistry. Here "No variation in chemistry" means no interannual variation in chemistry. The sentence is revised to clarify the meaning.*

l. 170. The results of using different resolutions and meteorological drivers, needs to be somehow included in the discussion of uncertainties.
*Thanks. Added in the section of Discussion and Conclusions.*

l. 194 here the operational definition of troposphere needs to be given.

*Thanks. Added.*

l. 202-204 In this case North American ozone is both from natural and anthropogenic emissions as well as stratospheric ozone that entered the North American region?

*Thanks. For a specific region, ozone produced in the troposphere over that region is named after that region, such as 'North American ozone'. Ozone produced in the stratosphere and then entered the troposphere is labelled as 'stratospheric ozone', an independent tracer. So, North American ozone excludes stratospheric ozone that entered North America. The original Table 2 (present Table S1) explained this. We further clarified this in this revision.*

l. 330 I recommend also to consider the results of Turnock et al, ACP, 2018, which discusses HTAP2 results.

*Thanks. Added.*

l. 346. This finding warrant a bit more discussion, given the similarity of emissions but longer distance compared to Europe.

*Thanks. The expression has been revised. On annual average, North America and Europe contributes 5-13 ppbv (7-12%) and 5-7 ppbv (3-11%) to ozone in the East Asian middle and upper troposphere, respectively. The annual mean of North American ozone is higher than that of European ozone over East Asia at layers above 500 hPa.*

*To compare the two, we further conducted four sensitivity experiments. In two of simulations, biogenic emissions in North America and Europe were turned off, respectively. In another two simulations, lightning $NO_x$ emissions in North America and Europe were turned off respectively. The difference between the results from the*

*control experiment and those from these four sensitivity simulations are shown in Figure S4. It is demonstrated that the difference between North America and European ozone over East Asia is from both anthropogenic and natural emissions including biogenic and lightning sources. The analysis has been added in the first paragraph of section 3.2.1.*

l. 368 I guess that can well be, also given the fact that many other models have larger stratospheric influxes. Would the conclusions change substantially if the number would be double. It is unclear to me how stratospheric ozone influx is accounted in the various attribution methods (e.g. influx of ozone In Europe, is that European ozone as well?)

*Thanks for the points. In this study, stratospheric ozone is ozone produced in the stratosphere and then transported into the troposphere. Stratospheric-to-troposphere ozone flux is the amount of the stratospheric ozone that entered the troposphere. European ozone is the ozone produced in the European troposphere. So, stratospheric-to-troposphere ozone flux over Europe is excluded in European ozone. Stratospheric ozone transported downward to the troposphere is showed in Figures 9 and 10. The values of ozone concentrations from different source regions (Figures 9a-9d) would not changed if the stratospheric ozone is doubled. We assessed the fractional contribution of ozone from different source regions in Figures 9e-9h and Figures 9i-9l, respectively. A change in the influxes of stratospheric ozone would not affect the results in Figures 9e-9h but in Figures 9i-9l. This is explained in section 2 in this revision.*

l. 410 It is not so clear why you would need 3x2 monsoon indices. If they are so different, please summarize what aspects they would represent stronger. Please also clarify how monsoon (summer phenomenon) indices also can have significant correlation with winter ozone.

*Thanks. We used three monsoon indices for winter and three for summer, respectively. In each season, the three indices describe the features of the EAM from different perspectives. The monsoon indices were correlated with ozone variation in the same season. The monsoon indices in summer were not connected to wintertime ozone. The explanation has been added. We have also added the correlation coefficients for each of the indices in Table S2, and the corresponding analysis is added in section 4.*

l. 540 It would have been great to use the HTAP2 compilation for 2008/2010- as did many other models. It would be good to mention at least the differences.

*Thanks. In future studies, the HTAP2 emission inventory can improve the estimates of foreign influences on ozone over East Asia and their long-term trends. The difference between the anthropogenic emissions in this study and that for HTAP2 is briefly discussed in the last section. From 2000 to 2010 in EDGAR emission inventories, $NO_x$, CO, and NMVOCs respectively changed by 9.5%, -1.2%, and 5.2% globally, and the three species decreased across North America and Europe but increased in East Asia (Turnock et al., 2018).*

---

## Author Comment (AC2) · 23 Jul 2019

*We are very grateful to the reviewers for their valuable comments and suggestions, which have helped us greatly in improving our manuscript. We have addressed all the comments and revised the manuscript accordingly. The point-to-point responses are provided below in Italic.*

Anonymous Referee #1

General Comment

This paper examined the tropospheric ozone in East Asia in terms of the influence of various source regions, particularly focused on the regions outside East Asia, they called them as "foreign ozone". The topic of the paper is well within the scope of the journal. The model used and the methods to deduce the influence of each different source region are adequate and have been applied so far in various studies for similar purpose. The results shown in the paper are generally consistent with the facts published in the previous literatures. This gives a certain reliability to the analysis done in this paper, but at the same time, the novelty of this paper over those previous studies is not clearly shown in the manuscript. For example, the authors reviewed the roles of East Asian Monsoon on the foreign ozone in East Asia in "Introduction", but the findings of this paper written in "Abstract" is quite similar to what was reviewed there. The author argued that this paper provided a comprehensive assessment of the influence of foreign ozone on the East Asian tropospheric ozone, but if they just want a comprehensive assessment, writing a review paper is more suitable, and actually the review given in this paper is quite comprehensive. I suggest the authors should state more clearly what they achieved on top of the previous studies to be published in the journal.

This is my general concern.

*Thanks for the reviewer's comments and suggestions. We added more statements in this revision in various sections on what is new in our study, mostly in the section of Discussion and Conclusions. This study reveals the significant foreign influences on*

*tropospheric ozone over East Asia through global atmospheric transport and gains some new insights. Firstly, this study comprehensively assessed foreign influences on tropospheric ozone over East Asia, while previous studies investigated ozone transport from one or a few foreign regions (X. Li et al., 2014; Chakraborty et al., 2015), that during one or a few seasons (Ni et al., 2018) or only surface ozone in East Asia (Wang et al., 2011). Comparisons with previous studies show that, despite some disagreements concerning various details, our results appear to be reasonable.*

*Secondly, we examined the foreign influence on ozone in East Asia throughout all tropospheric columns. The simulations show that the concentration of foreign ozone increases remarkably with altitude and is much higher than its native counterpart in the middle and upper troposphere. The influence in the East Asian middle and upper troposphere is important to climate change because of the considerable ozone radiative forcing over the area (Myhre et al., 2017). Such an impact has been rarely documented (Sudo and Akimoto, 2007).*

*Thirdly, we highlight the influences of EAM on the seasonal and interannual variations of foreign ozone distribution in East Asia, primarily through the vertical transport. Advancing from Zhu et al. (2017) for North American ozone, significant correlations between the strength of EAM and the ozone transport from various foreign regions including Europe, South Asia, and Southeast Asia have been found. These findings provide further understanding of the mechanisms of the intercontinental transport of air pollutants to East Asia.*

Major Comment:

Almost all analysis was done for the average over East Asia. As previous studies have shown that the relative contributions of various source regions can vary considerably depending on the location within East Asia. So, I cannot fully understand the meaning

of such "East Asian averaged" contributions of various source regions. Actually, the latitudinal dependence in each foreign and native contributions were analyzed, but I guess the longitudinal dependence should be also large enough to be analyzed.

*Thanks for the points. We have added two figures (Figures 3 and 8) to show how foreign ozone distributes horizontally over East Asia. The longitudinal variations of foreign ozone have been analysed as well (Figure S6).*

*.*

*Figure 3 compares the fractional contributions of native and foreign sources for ozone over East Asia in terms of the annual mean. The fractional contribution of foreign ozone at high altitudes is greater than that at the surface, and can reach a regional mean of up to 68% at 500 hPa (Figure 3f). At the surface, the fractional contribution of foreign ozone is lowest over South China, where it is lower than that of native ozone (Figure 3d and 3h). The analysis has been added in the second paragraph of section 3.1.*

*Figure 8 presents how foreign ozone from different source regions distributes horizontally in the middle troposphere, illustrating significant foreign impacts on ozone over East Asia at the altitudes. The streamlines in Figure 8 roughly show the transport pathways, demonstrating the importance of the westerlies in driving the ozone transport from North America, Europe, Africa, and central Asia to East Asia. The analysis has been added in the first paragraph of section 3.2.1.*

*The longitudinal variations of foreign ozone from different source regions over East Asia are shown in Figure S6. In the East Asian middle and upper troposphere, they are less obvious than the latitudinal variations, especially in winter (Figure 10 vs. Figure S6). In the East Asian middle troposphere (500 hPa), central Asian and South Asian ozone decreases with longitude in summer, varying insignificantly with longitude in winter. At the East Asian surface, ozone from the two regions decreases*

*with longitude. Longitudinal variations of ozone from North America, Europe, Africa, and Southeast Asia are less obvious than that for central Asia and South Asia. The analysis has been added in the last paragraph of section 3.2.3.*

Specific Comments:

- L45: The terms "native" and "foreign" are used before their definition is given in Table 2.

*Thanks. Table 2 is moved into the supplement as Table S1. 'Foreign ozone' and 'native ozone' have been explained when the terms first appear.*

- L83-85: Is the "transport" itself associated with thing other than meteorology such as emission and/or chemistry?

*The 'transport' process itself is driven by the meteorology, specifically the atmospheric circulation or the wind field. The amount of ozone brought by the transport from the source region to the receptor region is related to the emissions and chemistry. Therefore, the influences of the transport on ozone over the receptor region is associated with all these factors. This sentence has been revised.*

-L124-125: How did you treat CH4 chemistry in the model? Is it fully represented?

*In the full chemistry simulation, the $CH_4$ concentrations were fixed throughout the troposphere to annual zonal mean values in four latitudinal bands (90N-30N, 30N-Eq., Eq.-30S, and 30S-90S). The global annual mean anthropogenic emissions of $CH_4$ were from EDGAR v4.2. In our sensitivity simulations, the effects of $CH_4$ emissions on anthropogenic ozone were excluded. This limitation has been added in the discussion section.*

- L132-136: I don't think these detailed definition of regions are necessary, since Figure 1 shows them visually and they can be found in its caption.

*Agree. The definition of regions is removed.*

- L146 (formula (1)): How is the consistency between CTRL-EAnth-GLO and the sum in the denominator of the first term? How are they close to each other? Should be explained somewhere in the text.

*In Equation (1), $\sum_{i=1}^{8}$ (CTRL - EAnth-$X_i$) is the sum of the ozone response to the 100% perturbation for each region, CTRL - EAnth-GLO is the ozone response to the 100% perturbation for the globe. In the East Asian troposphere, ozone concentration from $\sum_{i=1}^{8}$ (CTRL - EAnth-$X_i$) is 0-4 ppbv (0-20%) higher than that from CTRL - EAnth-GLO (Figure S1). Figure S1 and explanations are added in this revision.*

- L159-160: The production and loss of ozone can vary considerably with in a day, so I imagine using daily production and loss data should have bad consequences on the simulated tagged ozone concentration. Did you check the validity for using the daily values for them?

*Thanks for this point. Due to the diurnal variations of ozone and the nonlinearity in chemistry, using daily production and loss data is one of the uncertainties for the tagged ozone simulation. We have compared the simulations between the full chemistry run and the tagged ozone run to assess the overall uncertainty of tagged ozone simulation (Figures S2-S3). The results show that the difference of the ozone concentrations in the two simulations is within ±5%. Results from the two types of simulations were also compared in Han et al. (2018). We also listed this as one of the sources of uncertainties in the section of Discussion and Conclusions.*

- Table 2: I don't think Table 2 could effectively explain the different definition of the terminology used in the manuscript. For me, the caption of Table 2 is easier to understand what you want to explain than Table2 itself and the descriptions in the

main text (L194-210).

*Table 2 has been moved into the supplement (Table S1). The terms are explained at their first appearance or along with the experiment description.*

- L233-234: You can not compare the contribution of foreign ozone on whole East Asia and that on China

*Thanks for this point. The sentence is removed.*

- L248: How did you calculate the chemical lifetime? Explain it in the text.

*'The chemical lifetime' has been revised to 'the lifetime'. For each model grid in the boundary layer, the lifetime of ozone was calculated by the daily average dry deposition and chemical loss rate of ozone. For each grid in the free troposphere, only the chemical loss rate of ozone is used. The explanation is added in the revision.*

- L251-252: Only the dry deposition could be the cause of the difference? Are there any other causes which should be mentioned here?

*The dry deposition and active chemical reactions of ozone in the boundary layer are the two main reasons for the shorter lifetime of ozone at the surface than in the free troposphere. The sentence has been revised accordingly.*

- L267-269: This part should be more specific. Where is the East Asian trough? Which region of downdraft you are referring?

*The East Asian trough in the middle troposphere is an important feature of the EAWM system. It locates along ~130-140ºE in winter (see Figure 6a in Y. Zhu et al. (2017)). Downdrafts prevail behind the East Asian trough in winter from 100ºE to 140ºE. The locations of the East Asian trough and downdrafts are added in the text.*

- L284-287: This sentence is quite hard to understand logically. The difference

between foreign and foreign anthropogenic O3 is not mentioned in the previous sentences. I cannot understand what you want to mean here.

*Thanks. The sentence is removed.*

- L342: Why the contribution from Africa in winter can be so large? Why it can be larger than that from SAS SEAS where much closer to EAS.

*Thanks for this question. Ozone transport from Africa to the East Asian middle and upper troposphere is mainly driven by the Hadley circulation and the subtropical westerlies. Unlike South Asia and Southeast Asia, Africa covers areas in the Northern Hemisphere (NH) and the Southern Hemisphere, with around three-quarters of the continent in the tropics. In NH winter, there still exists strong convection over the intertropical convergence zone (ITCZ) in Africa. In NH winter, the ITCZ in Africa uplifts biogenic emissions to the middle and upper troposphere, where is with high lightning $NO_x$ emissions. Therefore, ozone concentrations in the African middle and upper troposphere are relatively high. Furthermore, African ozone in the middle and upper troposphere can be efficiently transported to East Asia by the subtropical westerlies. Compared with Africa, although South Asian and Southeast Asia are more close to East Asia, the atmospheric circulations are less favorable to the ozone transport to East Asia (Figure S5). See Han et al. (2018) for more specific analysis of ozone transport from Africa to East Asia. We added explanations on this in the third paragraph of section 3.2.1 in this revision.*

- L442 (formula(8)): What is U'850?

*Thanks. U'850 is the anomaly of the zonal wind at 850 hPa. The explanation is added.*

- L484-487: Where should I look at in Figure 3 and 7 to find these values? Is it annual evaluation or seasonal? This is a quite blur sentence.

*This sentence has been revised. The concentrations of foreign ozone and native ozone*

*is compared on the annual average in this sentence. Such a comparison in the four seasons is added in section 3.1.*

---

## Author Response (AR2)

*Dear Dr. Frank Dentener,*

*We thank the reviewers for the valuable comments that help us improve our manuscript. We have made our revision in response to all the comments. The point-to-point responses are provided. The comparison of our manuscript between this version and last version is also provided.*

*Thank you and best regards,*

*Jane Liu, Ph.D*

Anonymous Referee #1:

- Introduction is too long and some sentences are inconsistent to each other. I recommend the authors to review the introduction part again and shortened it.

*Thanks. We have revised and shortened the Introduction section.*

- Authors use the term "ozone" to mean "tropospheric ozone". I don't like this terminology. It may cause unnecessary confusion to the readers.

*Thanks for this point. We agree. We removed "tropospheric ozone is also termed as ozone in this paper" in the definition text and in Table S1 in this revision.*

-L275-277: I don't think this sentence is necessary.

*Deleted.*

-U'850: Anomaly from what?

*$U'_{850}$ in Equation (8) is the anomaly of $U_{850}$ from the long-term mean climatology. We clarified this in this revision. Please see Line 497.*

-L553: Table 2 doesn't say so. Contributions of CAS and ROW are as large as those of NAM and EUR.

*Thanks for pointing this out. The phrase is revised accordingly. Please see Lines 546-*

*548.*

- Table 2 and Table 3 should be more precisely explained.

+ Which experiments listed in Table 1 was used for each Tables? This information should clearly be stated in Table caption or manuscript.

*Thanks. The experiments used for Tables 2-3 are added in the captions.*

+ How did you deduce "Natural ozone"? There is no description on this.

*"Natural ozone" was defined in the text and in Table S1 in the last revision. It is defined as the sum of ozone produced in the troposphere from precursors with non-anthropogenic sources (termed as non-anthropogenic ozone) and ozone from the*

*stratosphere (termed as stratospheric ozone). In this revision, the term of "natural ozone" is replaced with these two terms, i.e. non-anthropogenic ozone and stratospheric ozone, in Table 3, Table S1 and in the relevant text to improve clarity.*

+ How did you deduce "total"? Is this just a sum of all foreign contributions?

*Yes, 'total' is just the sum of all foreign contributions. We added this explanation in a note for the tables. Please see Lines 953-954 and 961-962.*

[revised manuscript text omitted]

$\text{EAWMI1} = -\text{GPH}_{500}(30\text{-}45\ ^{\circ}\text{N}, 125\text{-}145\ ^{\circ}\text{E})$ (2)

$\text{EAWMI2} = \text{U}_{300}(27.5\text{-}37.5\ ^{\circ}\text{N}, 110\text{-}170\ ^{\circ}\text{E}) - \text{U}_{300}(50\text{-}60\ ^{\circ}\text{N}, 80\text{-}140\ ^{\circ}\text{E})$ (3)

$\text{EAWMI3} = \text{WS}_{850}(25\text{-}50\ ^{\circ}\text{N}, 115\text{-}145\ ^{\circ}\text{E})$ (4)

$\text{EASMI1} = \text{U}_{850}(5\text{-}15\ ^{\circ}\text{N}, 90\text{-}130\ ^{\circ}\text{E}) - \text{U}_{850}(22.5\text{-}32.5\ ^{\circ}\text{N}, 110\text{-}140\ ^{\circ}\text{E})$ (5)

$\text{EASMI2} = \delta(10\text{-}40\ ^{\circ}\text{N}, 110\text{-}140\ ^{\circ}\text{E}, 850\ \text{hPa})$ (6)

[revised manuscript text omitted]